# Risk-Aware Transfer in Reinforcement Learning using Successor Features

**Michael Gimelfarb**\*
University of Toronto
mike.gimelfarb@mail.utoronto.ca

**André Barreto**
DeepMind
andrebarreto@google.com

**Scott Sanner**\*
University of Toronto
ssanner@mie.utoronto.ca

**Chi-Guhn Lee**
University of Toronto
cglee@mie.utoronto.ca

## Abstract

Sample efficiency and risk-awareness are central to the development of practical reinforcement learning (RL) for complex decision-making. The former can be addressed by transfer learning and the latter by optimizing some utility function of the return. However, the problem of transferring skills in a risk-aware manner is not well-understood. In this paper, we address the problem of risk-aware policy transfer between tasks in a common domain that differ only in their reward functions, in which risk is measured by the variance of reward streams. Our approach begins by extending the idea of generalized policy improvement to maximize entropic utilities, thus extending policy improvement via dynamic programming to sets of policies *and* levels of risk-aversion. Next, we extend the idea of successor features (SF), a value function representation that decouples the environment dynamics from the rewards, to capture the variance of returns. Our resulting risk-aware successor features (RaSF) integrate seamlessly within the RL framework, inherit the superior task generalization ability of SFs, and incorporate risk-awareness into the decision-making. Experiments on a discrete navigation domain and control of a simulated robotic arm demonstrate the ability of RaSFs to outperform alternative methods including SFs, when taking the risk of the learned policies into account.

## 1 Introduction

*Reinforcement learning* (RL) is a general framework for solving sequential decision-making problems, in which an agent interacts with an environment and receives continuous feedback in the form of rewards. However, many classical algorithms in RL do not explicitly address the need for *safety*, making them unreliable and difficult to deploy in some real-world applications [10]. One reason for this is the relative *sample inefficiency* of model-free RL algorithms, which often require millions of costly or dangerous interactions with the environment or fail to converge altogether [44, 46]. Transfer learning addresses these problems by incorporating prior knowledge or skills [23, 41]. Despite this, using the expected return as a measure of optimality could still lead to undesirable behavior such as excessive risk-taking, since low-probability catastrophic outcomes with negative reward and high variance could be underrepresented [29]. For this reason, risk-awareness is becoming an important aspect in the design of practical RL [14]. Thus, an ultimate goal of developing reliable systems should be to ensure that they are both sample efficient and risk-aware.

---

\*Affiliate to Vector Institute, Toronto, Canada.

35th Conference on Neural Information Processing Systems (NeurIPS 2021).

|  | Transfers Skills | Exploits Task Structure | Risk-Sensitive |
|---|:---:|:---:|:---:|
| RL [7, 11, 20, 27, 28, 30, 36, 39, 45] | ✗ | ✗ | ✓ |
| Transfer [15, 17, 19, 25, 26, 38, 43] | ✓ | ✗ | ✓ |
| Successor Features [2–4, 8] | ✓ | ✓ | ✗ |
| **RaSF (Ours)** | ✓ | ✓ | ✓ |

Table 1: Comparison of RaSF with relevant work in transfer learning and risk-aware RL.

We take a step in this direction by studying the problem of risk-aware policy transfer between tasks with different goals. A powerful way to tackle this problem in the risk-neutral setting is the *GPI/GPE* framework, of which *successor features* (SF) are a notable example [2]. Here, *generalized policy improvement* (GPI) provides a theoretical framework for transferring policies with monotone improvement guarantees, while *generalized policy evaluation* (GPE) facilitates the efficient evaluation of policies on novel tasks and is a key component in satisfying the assumptions of GPI in practice. Together, GPI/GPE provide strong transfer benefits in novel task instances even before any direct interaction with them has taken place, a phenomenon we call *task generalization*. The key to the superb generalization of GPI/GPE lies in their ability to directly exploit the structure of the task space, taking advantage of subtle differences and commonalities between task goals to transfer skills seamlessly in a *composable* manner. This property could be an effective way of tackling problems in *offline RL* [24], such as the transfer of skills learned in a simulator to a real-world environment. However in many cases, such as helicopter flight control [16], making one wrong decision could lead to catastrophic outcomes. Hence, being risk-aware could offer one way to avoid worst-case outcomes when transferring skills in real-world settings.

**Contributions.** We contribute a novel successor feature framework for transferring policies with the goal of maximizing the entropic utility of return in MDPs (Section 2.2). Intuitively, the entropic utility encourages agents to follow policies with predictable and controllable returns characterized by low variance, thus providing a natural way to incorporate risk-awareness. Furthermore, while our theoretical framework could be extended to other classes of utility functions, the entropic utility has many favorable mathematical properties [13, 22] that we exploit directly in this work to achieve *optimal* transfer (Lemma 1, Theorem 1 and 2). We also derive a form of risk-aware GPE based on the mean-variance approximation, in which the sufficient statistics of the return distribution can be computed directly (Section 3.3) or by leveraging recent developments in distributional RL [6]. Our resulting approach, which we call *Risk-Aware Successor Features* (RaSF), is able to exploit the task structure to achieve task generalization with respect to novel goal instances as well as levels of risk aversion, where emphasis is placed on avoiding high volatility of returns. Our approach is also complementary to other advances in successor features, including feature learning [3], universal approximation [8], exploration [21], and non-stationary reward preferences [4]. Empirical evaluations on discrete navigation and continuous robot control domains (Section 4) demonstrate the ability of RaSFs to better manage the trade-off between return and risk and avoid catastrophic outcomes, while providing excellent generalization on novel tasks in the same domain.

**Related Work.** The entropic and mean-variance objectives are popular ways of incorporating risk-awareness in RL [7, 11, 20, 27, 28, 30, 36, 39, 45]. However, transferring learned skills between tasks while taking risk into account is a difficult problem. One way to implement risk-aware transfer is to learn a critic [38] or teacher [43] that can guide an agent toward safer behaviors on future tasks. The risk-aware transfer of a policy from a simulator to a real-world setting has also been studied in the area of robotics [17]. Another approach for reusing policies is the *probabilistic policy reuse* of García and Fernández [15], but requires strong assumptions on the task space. *Hierarchical RL* (HRL) is another related approach that relies on hierarchical abstractions, enabling an agent to decompose tasks into a hierarchy of sub-tasks, and facilitating the transfer of temporally-extended skills from sub-tasks to the parent task. The *CISR* approach of Mankowitz et al. [26] is the first to investigate safety explicitly within HRL, followed up by work on *safe options* [18, 19, 25]. However, none of the existing work takes advantage of the compositional structure of task rewards to transfer skills while optimizing the variance-adjusted return, which is the problem we tackle in this paper (see Table 1).

## 2 Preliminaries

### 2.1 Markov Decision Process

Sequential decision-making in this paper follows the *Markov decision process* (MDP), defined as a four-tuple $\langle \mathcal{S}, \mathcal{A}, r, P \rangle$: $\mathcal{S}$ is a set of states; $\mathcal{A}$ is a finite set of actions; $r : \mathcal{S} \times \mathcal{A} \times \mathcal{S} \to \mathbb{R}$ is a bounded reward function, where $r(s, a, s')$ is the immediate reward received upon transitioning to state $s'$ after taking action $a$ in state $s$; and $P : \mathcal{S} \times \mathcal{A} \times \mathcal{S} \to [0, \infty)$ is the transition function for state dynamics, where $P(s'|s, a)$ is the probability of transitioning to state $s'$ immediately after taking action $a$ in state $s$.

In the episodic MDP setting, decisions are made over a horizon $\mathcal{T} = \{0, 1, \ldots T\}$ where $T \in \mathbb{N}$. We define a *stochastic Markov policy* as a mapping $\pi : \mathcal{S} \times \mathcal{T} \to \mathscr{P}(\mathcal{A})$, where $\mathscr{P}(\mathcal{A})$ denotes the set of all probability distributions over $\mathcal{A}$. Similarly, a *deterministic Markov policy* is a mapping $\pi : \mathcal{S} \times \mathcal{T} \to \mathcal{A}$. In the *risk-neutral* setting, the goal is to find a policy $\pi$ that maximizes the expected sum of future rewards after initially taking action $a$ in state $s$,

$$Q_h^\pi(s, a) = \mathbb{E}_{s_{t+1} \sim P(\cdot | s_t, a_t)} \left[ \sum_{t=h}^{T} r(s_t, a_t, s_{t+1}) \,\Big|\, s_h = s, \, a_h = a, \, a_t \sim \pi_t(s_t) \right].$$

In this case, it is possible to show that a deterministic Markov policy $\pi^*$ is optimal [33]. The theoretical framework in this paper also allows for time-dependent reward or transition functions.

### 2.2 Entropic Utility Maximization

We incorporate risk-awareness into the decision-making by maximizing the *entropic utility* $U_\beta$ of the cumulative reward, defined for a fixed $\beta \in \mathbb{R}$ as

$$U_\beta[R] = \frac{1}{\beta} \log \mathbb{E} \left[ e^{\beta R} \right], \tag{1}$$

for real-valued random variables $R$ on a bounded support $\Omega \subset \mathbb{R}$. An important property of the entropic utility is the Taylor expansion $U_\beta[R] = \mathbb{E}[R] + \frac{\beta}{2} \mathrm{Var}[R] + O(\beta^2)$. Interpreting the risk as return variance, $\beta$ can now be interpreted as the risk aversion of the agent: choosing $\beta < 0$ ($\beta > 0$) leads to *risk-averse* (*risk-seeking*) behavior, while $\beta = 0$ is *risk-neutral*, e.g. $U_0[R] = \mathbb{E}[R]$.

Specializing (1) to the MDP setting, the goal is to maximize

$$\mathcal{Q}_{h,\beta}^\pi(s, a) = U_\beta \left[ \sum_{t=h}^{T} r(s_t, \pi_t(s_t), s_{t+1}) \right] \tag{2}$$

over all policies starting from $s_h = s$ and $a_h = a$. As in the risk-neutral setting, it is possible to show that a deterministic Markov policy is optimal [5]. Furthermore, $\mathcal{Q}_{h,\beta}^\pi$ can be computed iteratively through time using the *Bellman equation* [9, 32]:

$$\begin{aligned}
\mathcal{Q}_{h,\beta}^\pi(s, a) &= U_\beta \left[ r(s, a, s') + \mathcal{Q}_{h+1,\beta}^\pi(s', \pi_{h+1}(s')) \right] \\
&= \frac{1}{\beta} \log \mathbb{E}_{s' \sim P(\cdot | s, a)} \left[ \exp \left\{ \beta \left( r(s, a, s') + \mathcal{Q}_{h+1,\beta}^\pi(s', \pi_{h+1}(s')) \right) \right\} \right],
\end{aligned} \tag{3}$$

starting with $\mathcal{Q}_{T+1,\beta}^\pi(s, a) = 0$. In fact, (1) is the *only* utility function that has this equivalence, and other key properties (Lemma 1), while also satisfying *time consistency* that ensures the learned risk-aware behaviors remain consistent across time [22]. In this paper, we use (3) to establish a general GPI framework for risk-aware transfer learning with provable guarantees, and leverage approximations of (2) to learn portable policy representations.

In reinforcement learning, the Bellman equation is not applied directly since it suffers from the curse of dimensionality when $\mathcal{S}$ is high-dimensional or continuous, and since neither the dynamics nor the reward function are often known. Instead, the agent interacts with the environment using a stochastic exploration policy $\pi^e$, collects trajectories $\{(s_t, a_t, s_{t+1}, r_{t+1})\}_{t=0}^{T-1}$, and updates $\mathcal{Q}_{h,\beta}(s_t, a_t)$ via sample approximations $\hat{U}_\beta \approx U_\beta$ [36]. Our goal is to ameliorate the relative sample-inefficiency of RL through transfer learning that we aim to generalize to the risk-aware setting.

## 2.3 Transfer Learning

We now formalize the general *transfer learning* problem. Let $\mathcal{M}$ be the set of all MDPs with shared transition function $P$ but different (bounded) reward functions. A fixed set of source tasks $M_1, \ldots M_n \in \mathcal{M}$ is instantiated, and their corresponding optimal policies $\pi_1, \ldots \pi_n$ are estimated. Our main goal is to transfer these resulting source policies to a new target task $M_{n+1} \in \mathcal{M}$, to obtain a policy $\pi_{n+1}^*$ whose utility is better than one learned from scratch using only a fixed number of samples from $M_{n+1}$. We refer to this outcome as *positive transfer*.

As discussed earlier, a standard way to implement transfer learning is the GPI/GPE framework of Barreto et al. [2]. The core mechanism that enables positive transfer in the risk-neutral setting is called *generalized policy improvement* (GPI). Specifically, the set of source policies $\pi_1, \ldots \pi_n$ are evaluated on the target task $M_{n+1}$ to obtain corresponding values $Q_{n+1}^{\pi_1}, \ldots Q_{n+1}^{\pi_n}$. Given a mechanism that can perform this policy evaluation step efficiently with some small error $\varepsilon$ — namely successor features discussed and extended in Section 3.3 — an agent then selects actions in a greedy manner by following policy $\pi(s) \in \arg\max_a \max_{j=1\ldots n} Q_{n+1}^{\pi_j}(s, a)$ in state $s$. The policy $\pi$ corresponds to a strict policy improvement operator, and thus fulfills our requirements for positive transfer.

# 3 Risk-Aware Transfer Learning

An obvious challenge of applying GPI in the risk-aware setting is that transferring optimal risk-neutral source policies does not guarantee risk-aware optimality in the target task. A much stronger observation is that, even if the source policies $\pi_j$ are risk-aware, performing the policy evaluation step in a risk-neutral way can still break the risk-awareness of GPI. This makes the extension of GPI to the risk-aware setting a non-trivial problem.

## 3.1 A Motivating Example

To see this, consider the MDP shown in Figure 1, which involves navigating from an initial state 'S' to a goal state 'G' in a grid with stochastic transitions. Traps of two types (X, Y) are placed in fixed cells, which upon entry terminate the episode with fixed costs, summarized compactly as pairs $(c_1, c_2)$. We define two source tasks with costs $(20, 20)$ and $(0, 0)$ and a target task with costs $(20, 0)$. The optimal policies for $\beta = -0.1$ induce three different trajectories: safe (blue) and hazardous routes (red) for the source tasks, and a relatively safe route (green) for the target task. We also note that risk-awareness is prerequisite for this problem, since a risk-neutral agent prefers the hazardous path when optimizing any of the tasks.

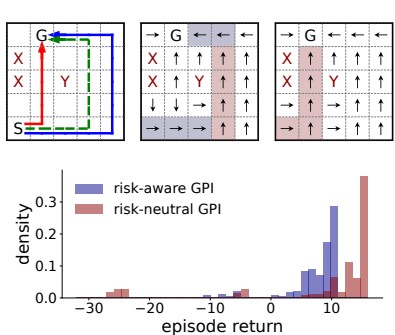

Figure 1: Comparing risk-aware and risk-neutral GPI. The risk-aware (for $\beta = -0.1$) GPI policy is shown in the top-middle plot, while the risk-neutral (for $\beta = 0$) GPI policy is shown in the top-right plot.

We compute the GPI policies $\pi$ with risk-averse ($\beta = -0.1$) and risk-neutral ($\beta = 0$) policy evaluation, as shown in the middle and rightmost plots in the top row in Figure 1. The bottom plot shows the distribution of returns collected over 5,000 test runs by acting according to the two GPI policies. Risk-averse policy evaluation results in an optimal risk-averse GPI policy corresponding to the green trajectory, whereas risk-neutral policy evaluation does not, even though both source policies are optimal on their corresponding tasks. Interestingly, the risk-averse GPI policy is a non-trivial stitching of the two source policies.

## 3.2 Risk-Aware Generalized Policy Improvement

Motivated by this example, we conjecture that the risk-awareness of the GPI policy $\pi$ is primarily dependent on the way in which the source policies are evaluated in target task instances. In this section, we describe theoretical results that generalize the concept of generalized policy iteration to the problem of maximizing the entropic utility of returns.

We begin by summarizing key properties necessary for establishing convergence of risk-aware GPI in the following lemma.

**Lemma 1.** *Let $\beta \in \mathbb{R}$ and $X, Y$ be arbitrary random variables on $\Omega$. Then:*

    *(1) (monotonicity) if $\mathbb{P}(X \geq Y) = 1$ then $U_\beta[X] \geq U_\beta[Y]$*
    *(2) (cash invariance) $U_\beta[X + c] = U_\beta[X] + c$ for every $c \in \mathbb{R}$*
    *(3) (convexity) if $\beta < 0$ ($\beta > 0$) then $U_\beta$ is a concave (convex) function*
    *(4) (non-expansion) for $f, g : \Omega \to \Omega$, it follows that*

$$|U_\beta[f(X)] - U_\beta[g(X)]| \leq \sup_{P \in \mathscr{P}_X(\Omega)} \mathbb{E}_P |f(X) - g(X)|,$$

*where $\mathscr{P}_X(\Omega)$ is the set of all probability distributions on $\Omega$ that are absolutely continuous w.r.t. the true distribution of $X$.*

A proof is provided in Appendix B.2. Properties (1)-(3) characterize *concave utilities* [13], which intuitively can be seen as minimal requirements for rational decision-making: (1) a lottery that pays off more than another in every possible state of the world should always be preferred; (2) adding cash to a position should not increase its risk; and (3) the overall utility can be improved by diversifying across different risks. Property (4) is a derivative of the first three, and thus the theoretical results in this work could be extended to the broader class of iterated concave utilities [34].

We can now state the first main result of the paper.

**Theorem 1 (GPI for Entropic Utility).** *Let $\pi_1, \ldots \pi_n$ be arbitrary deterministic Markov policies with approximate entropic utilities $\tilde{\mathcal{Q}}_{h,\beta}^{\pi_1}, \ldots \tilde{\mathcal{Q}}_{h,\beta}^{\pi_n}$ when evaluated in an arbitrary task $M$, with error $|\tilde{\mathcal{Q}}_{h,\beta}^{\pi_i}(s, a) - \mathcal{Q}_{h,\beta}^{\pi_i}(s, a)| \leq \varepsilon$ for all $s \in \mathcal{S}$, $a \in \mathcal{A}$, $i = 1 \ldots n$ and $h \in \mathcal{T}$. Define*

$$\pi_h(s) \in \arg\max_{a \in \mathcal{A}} \max_{i=1\ldots n} \tilde{\mathcal{Q}}_{h,\beta}^{\pi_i}(s, a), \quad \forall s \in \mathcal{S}. \tag{4}$$

*Then,*

$$\mathcal{Q}_{h,\beta}^{\pi}(s, a) \geq \max_i \mathcal{Q}_{h,\beta}^{\pi_i}(s, a) - 2(T - h + 1)\varepsilon, \quad h \leq T.$$

Thus, evaluating the risk of source policies using $U_\beta$ provides monotone improvement guarantees for GPI, and thus satisfies our definition of positive transfer. Another significant property of the bound in Theorem 1, and the one in Theorem 2 below, is the linear separation between the optimal utility and the approximation error $\varepsilon$. Knowing how the optimality of $\pi$ explicitly depends on $\varepsilon$, and how errors are propagated throughout the transfer learning process, is critical for developing reliable RL with predictable behavior, and highlights a key advantage of making GPI risk-aware. The additive relationship between the optimality and the approximation error in Theorem 1 further explains why SFs are robust to approximation errors. This becomes particularly advantageous in our setting, since estimating utilities $U_\beta$ accurately with GPE is more complicated than the risk-neutral setting.

A stronger result for GPI can be derived when the source policies $\pi_1, \pi_2 \ldots \pi_n$ are $\varepsilon$-optimal, and policy evaluation is once again performed using $U_\beta$. In this case, the optimality of GPI is determined by the similarity $\delta_r$ between the source and target task instances.

**Theorem 2.** *Let $\mathcal{Q}_{h,\beta}^{\pi_i^*}$ be the utilities of optimal Markov policies $\pi_i^*$ from task $M_i$ but evaluated in task $M$ with reward function $r(s, a, s')$. Furthermore, let $\tilde{\mathcal{Q}}_{h,\beta}^{\pi_i^*}$ be such that $|\tilde{\mathcal{Q}}_{h,\beta}^{\pi_i^*}(s, a) - \mathcal{Q}_{h,\beta}^{\pi_i^*}(s, a)| \leq \varepsilon$ for all $s \in \mathcal{S}$, $a \in \mathcal{A}$, $h \in \mathcal{T}$ and $i = 1 \ldots n$, and let $\pi$ be the corresponding policy in (4). Finally, let $\delta_r = \min_{i=1\ldots n} \sup_{s,a,s'} |r(s, a, s') - r_i(s, a, s')|$. Then,*

$$\left| \mathcal{Q}_{h,\beta}^{\pi}(s, a) - \mathcal{Q}_{h,\beta}^{*}(s, a) \right| \leq 2(T - h + 1)(\delta_r + \varepsilon), \quad h \leq T.$$

These results are proved in Appendix B.2 for the episodic setting and in Appendix B.3 for the discounted setting. Also, please note that these bounds are tight (see, e.g. Nemecek and Parr [31] for the risk-neutral setting). Finally, while not required in this work, the above results could be extended to more general settings in risk-averse control [34], though practical implementation of GPE in these settings remains an open problem.

## 3.3 Risk-Aware Generalized Policy Evaluation

Following Barreto et al. [2], let $\phi : \mathcal{S} \times \mathcal{A} \times \mathcal{S}' \to \mathbb{R}^d$ be a bounded and task-independent feature map, and consider the following linear representation of rewards,

$$r(s, a, s') = \phi(s, a, s')^\mathsf{T} \mathbf{w}, \quad \forall s, a, s',$$

where $\mathbf{w} \in \mathbb{R}^d$ is a task-dependent vector of reward parameters. The *risk-neutral* return becomes:

$$Q_h^\pi(s, a) = \mathbb{E}_P \left[ \sum_{t=h}^T \phi(s_t, a_t, s_{t+1})^\mathsf{T} \mathbf{w} \,\bigg|\, s_h = s, \, a_h = a, \, a_t \sim \pi_t(s_t) \right]$$

$$= \mathbb{E}_P \left[ \sum_{t=h}^T \phi(s_t, a_t, s_{t+1}) \,\bigg|\, s_h = s, \, a_h = a, \, a_t \sim \pi_t(s_t) \right]^\mathsf{T} \mathbf{w} = \psi_h^\pi(s, a)^\mathsf{T} \mathbf{w}, \quad (5)$$

where $\psi_h^\pi(s, a)$ are the *successor features* (SFs) associated with the policy $\pi$. The linear dependence of the return on $\mathbf{w}$ allows for instantaneous policy evaluation in novel tasks with arbitrary reward preferences $\mathbf{w}$, making it a particular — and perhaps the canonical — instantiation of GPE. More critically, $\psi_h^\pi$ can be seen as a *task-independent* and highly portable linear feature representation of policies, and it is the key to the generalization ability of SFs on novel task instances.

The concept of GPE can be generalized to incorporate entire distributions of the return. Repeating the above derivation for the entropic utility (2), we have:

$$\mathcal{Q}_{h,\beta}^\pi(s, a) = U_\beta \left[ \sum_{t=h}^T r(s_t, \pi_t(s_t), s_{t+1}) \right] = U_\beta \left[ \Psi_h^\pi(s, a)^\mathsf{T} \mathbf{w} \right], \quad (6)$$

corresponding to the random vector $\Psi_h^\pi(s, a) = \sum_{t=h}^T \phi_t$ of unrealized feature returns at time $h$. Thus, we have transformed the problem of estimating the utility of returns into the problem of estimating the distribution of $\Psi_h^\pi(s, a)^\mathsf{T} \mathbf{w}$. The key question now is how to estimate this distribution for fast GPE.

A natural way to do this is by applying a second-order Taylor expansion for $U_\beta$, since it allows us to precompute and cache the necessary moments of the return distribution:

$$U_\beta \left[ \Psi_h^\pi(s, a)^\mathsf{T} \mathbf{w} \right] = \mathbb{E}_P[\Psi_h^\pi(s, a)^\mathsf{T} \mathbf{w}] + \frac{\beta}{2} \mathrm{Var}_P[\Psi_h^\pi(s, a)^\mathsf{T} \mathbf{w}] + O(\beta^2)$$

$$\approx \psi_h^\pi(s, a)^\mathsf{T} \mathbf{w} + \frac{\beta}{2} \mathbf{w}^\mathsf{T} \mathrm{Var}_P[\Psi_h^\pi(s, a)] \mathbf{w} = \tilde{\mathcal{Q}}_{h,\beta}^\pi(s, a), \quad (7)$$

in which $\mathrm{Var}_P[\Psi_h^\pi(s, a)] = \Sigma_h^\pi(s, a)$ is interpreted as a covariance matrix for SFs. In the context of Theorems 1 and 2, the term $\varepsilon$ encapsulates the errors in the approximation of $\psi_h^\pi$ and $\Sigma_h^\pi$, plus the terms contained in $O(\beta^2)$ above. However, the main advantage of (7) is that, like (5), it is also analytic in $\mathbf{w}$ and allows for *instantaneous policy evaluation* with arbitrary reward preferences $\mathbf{w}$. Interestingly, (7) also allows instantaneous policy evaluation with respect to different choices of $\beta$, making it possible to revise or adapt the level of risk aversion on-demand. In all these cases, $\psi_h^\pi$ and $\Sigma_h^\pi$ provide task-independent and portable representations of policies while also accounting for exogenous risk. This is the key to preserving the task generalization ability of SFs in the risk-aware setting, and (7) can now be seen as a particular instantiation of GPE. We call this overall approach *Risk-aware Successor Features* (RaSF).

The simplest approaches for estimating $\Sigma_h^\pi$ in the exact (e.g. tabular) Q-learning setting are based on dynamic programming [37, 40], which would allow the overall approach to be easily integrated into existing SF implementations. In particular, the covariance satisfies the Bellman equation

$$\Sigma_h^\pi(s, a) = \mathbb{E}_{s' \sim P(\cdot|s,a)} \left[ \delta_h \delta_h^\mathsf{T} + \Sigma_{h+1}^\pi(s', \pi_{h+1}(s')) \,|\, s_h = s, \, a_h = a \right], \quad (8)$$

where $\delta_h$ are the Bellman residuals of $\psi_h^\pi(s, a)$. The approximation $\tilde{\psi}_h^\pi$ is known to converge to the true value $\psi_h^\pi$, and a similar result also holds for updating the covariance based on (8).

**Theorem 3 (Convergence of Covariance).** *Let $\| \cdot \|$ be a matrix-compatible norm, and suppose there exists $\varepsilon : \mathcal{S} \times \mathcal{A} \times \mathcal{T} \to [0, \infty)$ such that:*

1. $\|\tilde{\boldsymbol{\psi}}_h^\pi(s,a) - \boldsymbol{\psi}_h^\pi(s,a)\|^2 \le \varepsilon_h(s,a)$

2. $\|\mathbb{E}_{s' \sim P(\cdot|s,a)}[\tilde{\boldsymbol{\delta}}_h(\tilde{\boldsymbol{\psi}}_h^\pi(s',\pi_{h+1}(s')) - \boldsymbol{\psi}_h^\pi(s',\pi_{h+1}(s')))^\mathsf{T}]\| \le \varepsilon_h(s,a).$

*Then,*

$$\left\| \Sigma_h^\pi(s,a) - \mathbb{E}_{s' \sim P(\cdot|s,a)}\left[ \tilde{\boldsymbol{\delta}}_h \tilde{\boldsymbol{\delta}}_h^\mathsf{T} + \tilde{\Sigma}_{h+1}^\pi(s',\pi_{h+1}(s')) \right] \right\| \le 3\varepsilon_h(s,a).$$

A proof can be found in Appendix B.4. Appendix A.1 describes how $\tilde{\boldsymbol{\psi}}_h^\pi$ and $\tilde{\Sigma}_h^\pi$ can be learned online from environment interactions, while Appendix A.4 discusses further generalizations of (7). There are, however, several limitations of estimating $\Sigma_h^\pi$ in this way. First, obtaining accurate estimates of $\Sigma_h^\pi$ requires accurate estimates of $\boldsymbol{\psi}_h^\pi$ (thus estimating one quantity on top of another), making this approach difficult to apply with deep function approximation. This claim is further substantiated by Theorem 3 and preliminary experiments. A second issue that occurs is *double sampling*, when the same transitions are used to update the mean and covariance, resulting in accumulation of bias in the latter [1, 47]. Our experiments on the reacher domain mitigate these issues by leveraging distributional RL to approximate (7), while maintaining computational efficiency.

## 4    Experiments

To evaluate the performance of RaSF, we revisit the benchmark domains in Barreto et al. [2], which have been slightly modified for learning and evaluating risk-aware behaviors. We defer all experimental details to Appendix C.

### 4.1    Four-Room

**Domain Description.**    The first domain consists of a family of navigation tasks defined on a discrete 2-D space divided into four rooms, as illustrated on the left in Figure 2. The environment has additional objects that can be picked up by the agent by occupying their cells. These objects belong to one of three possible classes, drawn as different shapes in Figure 2, which determine their reward. The position of the objects remains fixed, but the rewards of their classes are reset every 20,000 transitions to random values sampled uniformly in $[-1,+1]$. To incorporate risk, traps are placed in fixed cells marked with X. For every time instant during which the agent occupies a trap cell, the trap activates spontaneously with a small probability, resulting in an immediate penalty and termination of the episode (we refer to this event as a failure). The goal is to maximize the total reward accumulated over 128 random task instances while minimizing the number of failures.

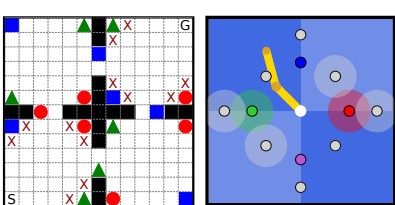

Figure 2: **Four-Room (Left):** the shapes of the objects represent their classes, 'S' is the start state, 'G' is the final goal state, and 'X' is a trap. **Reacher (Right):** colored and gray circles represent training and test targets, respectively, while shaded regions represent areas of high risk.

**Baselines.**    In order to demonstrate the power of our approach in the absence of approximation errors, we define a simple instance of RaSFs in which $\boldsymbol{\psi}^{\pi_i}$ and $\Sigma^{\pi_i}$ are learned exactly using lookup tables and dynamic programming (equation (8)). We also apply modest discounting of rewards to ensure that the Q-function converges, as is standard in RL and discussed further in Appendix A.2. The vector **w** is also learned using immediate reward feedback and exact, sparse state features $\phi$ provided to the agent. Due to its similarity to standard Q-learning, we call this approach RaSFQL. To provide a challenging baseline for comparison, we implemented another policy reuse algorithm (PRQL) [12]. Further replacing the risk-neutral action selection mechanism of PRQL with *smart exploration* [16] allows PRQL to easily account for return volatility, and we refer to this baseline as RaPRQL.

**Main Results.**    The performance of these algorithms is shown in Figure 3. The cumulative reward obtained by RaSFQL is generally lower than SFQL, as expected since a risk-averse agent should avoid the objects in the bottom-left and top-right rooms and forgo their associated rewards. Interestingly, the performance of RaSFQL far exceeds that of RaPRQL and even PRQL, suggesting that the benefits

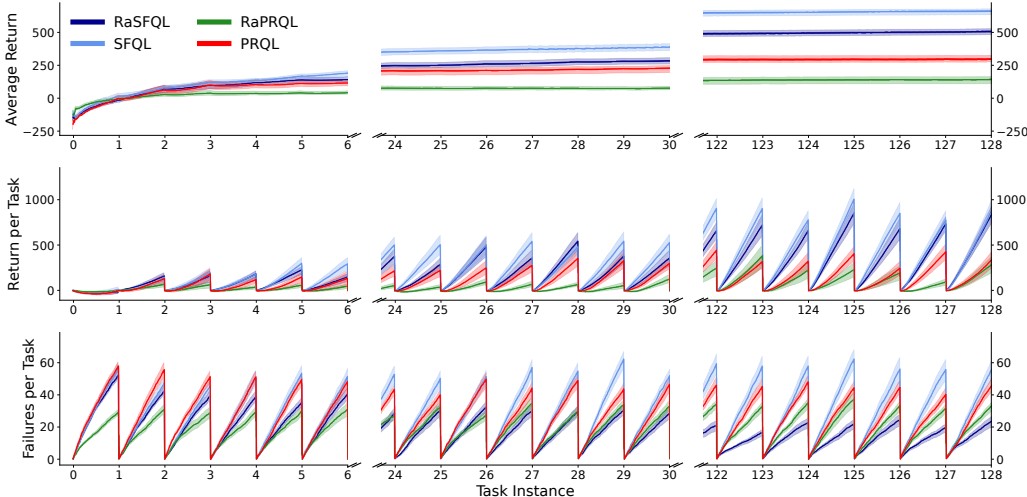

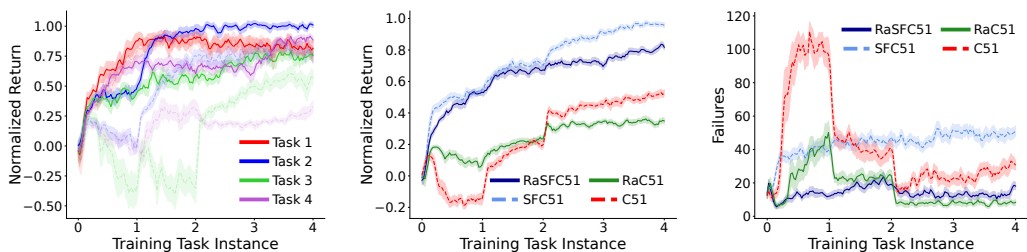

Figure 3: Average return, cumulative return and number of failures per task in the four-room domain, for $\beta = \omega = -2$. Shaded bands show one standard error over 30 independent runs.

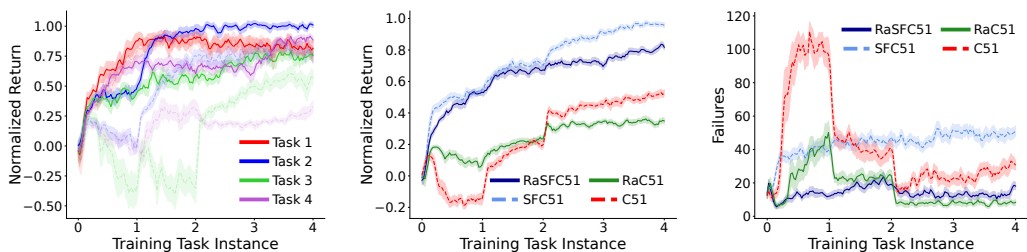

Figure 4: Performance on training and test tasks for the reacher domain, as a function of the number of experiences collected from the training tasks. **Left:** Normalized return on training tasks. Faded curves correspond to C51 performance. **Middle:** Normalized return averaged across all test tasks. **Right:** Total failures across all test tasks. Shaded bands show one standard error over 10 independent runs with different seeds.

of task generalization provided by GPI/GPE are quite strong. Furthermore, the number of failures observed by RaSFQL gradually decreases over the task instances, while the number of failures of SFQL slightly increases. This is consistent with Theorem 1 that guarantees monotone improvement in the *risk-adjusted* return of RaSFQL. On the other hand, while RaPRQL also learns to avoid risk, it fails slightly more often than RaSFQL. This suggests that the benefits of task generalization promised by GPI/GPE even allow risk-aware behaviors to emerge sooner than by using generic policy reuse methods that are unable to exploit the task structure, namely PRQL. This aspect becomes critical for minimizing failures when deploying a trained policy library on novel task instances in a real-world setting. Further analysis and ablation studies are provided in Appendix D.1.

### 4.2 Reacher

**Domain Description.** The second domain consists of a set of tasks based on the MuJoCo physics engine [42] that involve the maneuver of a robotic arm toward a fixed target location. As illustrated in the rightmost plot in Figure 2, the agent is only allowed to train on 4 tasks, whose target locations are indicated by colored circles, and must be able to perform well on 8 test tasks whose target locations are indicated by the grey circles. Furthermore, we incorporate two sources of reward volatility: (1) actions are perturbed by additive Gaussian noise; and (2) fixed regions around some of the target locations randomly incur negative rewards (failures), illustrated by faded circles in Figure 2. Please note that most of these high-volatility regions are centered on target locations of test tasks from which

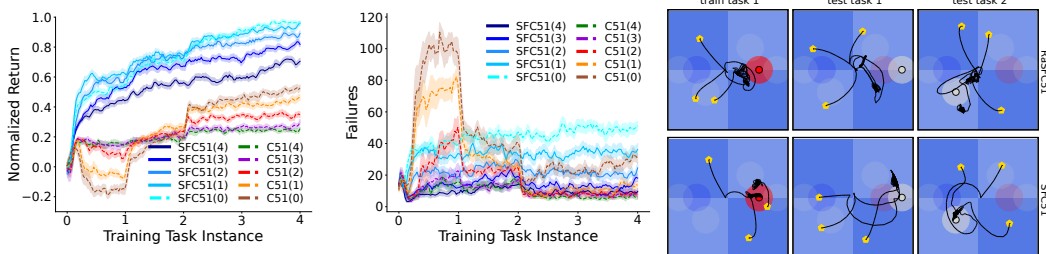

Figure 6: **Left:** Normalized average test return for the reacher domain, for different values of $\beta$ (legend values indicate the negative values of $\beta$). **Middle:** Total failures across all test tasks for various values of $\beta$ (legend values indicate the negative values of $\beta$). **Right:** Evolutions of the arm tip position during three successful rollouts of the reacher domain according to the GPI policy obtained after training with $\beta = -3$ (yellow pentagons indicate the initial states in each rollout). Only one training task and two test tasks are shown. The risk-averse agent learns to hover close to the goal while avoiding the high-volatility shaded regions.

the agent never learns directly. This stresses the agent's ability to avoid unforeseen dangers in the environment, in additional to performing well on previously unseen task instances.

**Baselines.** As discussed earlier, it is difficult to compute $\Sigma^{\pi_i}$ directly using (8). A computationally tractable way to avoid these issues is to first approximate the density of $\Psi^{\pi_i}(s, a)$, and then extract the moments needed to compute (8). Specifically, we apply C51 [6] by modeling $\Psi_1^{\pi_i}(s, a), \ldots \Psi_d^{\pi_i}(s, a)$ using histograms for each $(s, a)$. However, $\Psi^{\pi_i}$ are high-dimensional, so we avoid the curse of dimensionality by modeling the marginals $\Psi_j^{\pi_i}$ rather than their full joint distribution. This still turns out to be an effective way of detecting high-variance scenarios in the environment. The final architecture is illustrated in Figure 5,

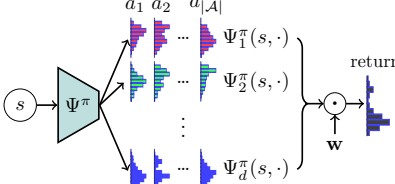

Figure 5: Architecture for $\Psi^\pi(s, a)$.

where the marginal distributions of $\Psi^{\pi_i}$ are modeled as separate output "heads" with a shared state encoder. The rest of the training protocol is identical to the SFDQN of Barreto et al. [2], except that DQN is replaced by the C51 architecture above, as further detailed in Appendix A.3. The risk-averse and risk-neutral instances of this approach for modeling successor features are referred to as RaSFC51 and SFC51, respectively, while RaC51 and C51 replace successor features with *universal value functions* [35] for generalization across target locations.

**Main Results.** The performance of the algorithms on the reacher domain is illustrated in Figure 4. We first observe that the performance of all four training policies for RaSFC51 improves almost immediately, obtaining returns that exceed 75% of the performance of a *fully trained* C51 agent, an observation that correlates highly with the original SFDQN. Interestingly, RaC51 is unable to achieve satisfactory performance on two out of the four training tasks, even after fully trained on all 4 tasks. Furthermore, the performance of RaSFC51 on the testing tasks far exceeds that of both RaC51 and C51, demonstrating the superior generalization ability of SFs in the risk-aware setting. Finally, the total number of failures across the test task instances is also considerably lower for RaSFC51 than it is for SFC51, and remains low during the entire training horizon. RaC51 and C51 fail frequently at the beginning of training, but less often later in training. While this may suggest that C51 is learning adequate risk-sensitive policies, it is mainly due to the fact that C51 is not able to generalize nearly as well as SFC51 for some of the test target locations, as elaborated in Appendix D.2.

**Additional Analysis.** We also conducted a sensitivity analysis for the risk-aversion parameter $\beta$ for each of the algorithms, summarized in Figure 6. The performance of RaSFC51 and RaC51 decays gracefully as the level of risk-aversion $\beta$ is increased in magnitude, but the performance of RaSFC51 uniformly outperforms RaC51 for *every* value of $\beta$ tested. The middle plot demonstrates that the number of failures also decreases drastically as $\beta$ is increased in magnitude, which stabilizies around $\beta = -4$. Furthermore, while C51 and RaC51 generally fail less often than SFC51 and RaSFC51, this

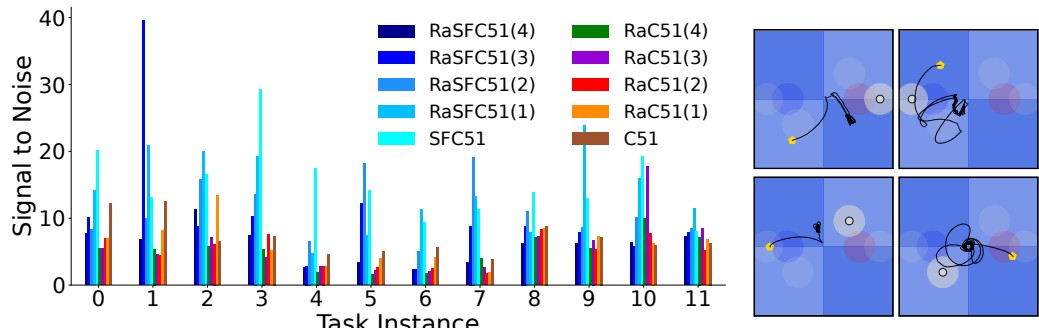

Figure 7: **Left:** Empirical signal-to-noise ratios of the GPI policies at the end of training. **Right:** Examples of worst-case behavior observed by RaSFC51 with $\beta = -3$, which depict the most likely sources of high return variability. While the agent learns to successfully avoid the risky regions of the domain, high return variance between rollouts can sometimes still be accumulated in other ways, such as failure to find a globally optimal policy (top left), inability to generalize the utility precisely on an unseen task for all states (bottom left), policy instability (top right) and divergence in the most extreme cases (bottom right) due to compounding of numerical errors.

is due to their inability to generalize their learned behavior to all novel target instances, as mentioned above. The final plot on the right shows that RaSFC51 indeed learns safer policies than SFC51 on both training and test instances. Specifically, RaSFC51 learns to hover as close to the goal as possible in most cases, while still avoiding the high-volatility shaded regions.

**Failure Modes and Limitations.** Finally, we compared the empirical signal-to-noise ratios[2] achieved by each baseline on each task using the returns of sampled trajectories at the end of training, and the results are summarized in Figure 7. RaSFC51 and SFC51 soundly outperform RaC51 and C51, and RaSFC51 generally obtains the highest ratios for medium values of $\beta \in \{-3, -2, -1\}$. However, RaSFC51 does not consistently outperform SFC51 on all domains, despite RaSFC51 avoiding the high-risk regions of the domain. We suspect that return variance is accumulated in other ways, some of which can be identified by examining the worst-case behaviors exhibited by the risk-aware GPI policy illustrated in the rightmost plots of Figure 7. However, we believe that these results are expected. Firstly, estimation of only the expected return in an off-policy setting with bootstrapping and deep function approximation is already a difficult problem [44], and addressing it is outside the scope of this work. Secondly, generalizing the empirical distribution learned on one set of tasks to a completely new set of *unseen* tasks – even within the same domain – is an even more challenging problem. We believe that advances in distributional RL that can learn more accurate distributions will strongly benefit our approach, as will methods suited for estimating the joint distributions of correlated returns efficiently.

## 5   Conclusion

We presented Risk-aware Successor Features (RaSFs) for realizing policy transfer between tasks with shared dynamics, with the goal of maximizing the entropic utility of return. We extended GPI to the risk-aware setting, providing monotone convergence and optimality guarantees, assuming that the utility of source policies can be evaluated. To facilitate policy evaluation, we also extended the notion of GPE to the risk-aware setting. Together, risk-aware GPI and GPE inherit the superior task generalization abilities of successor features, while learning to avoid dangerous high-volatility regions. More generally, incorporating risk and safety in sequential decision-making is a complex problem. The entropic utility objective does not capture tail risk nor other properties of the return distribution, which could be a challenging but powerful extension of successor features.

---

[2]This is computed as $\hat{\mu}/\hat{\sigma}$, where $\hat{\mu}$ is the estimated return and $\hat{\sigma}$ is the estimated standard deviation of the return accumulated across multiple rollouts. We use the signal-to-noise ratio because it is independent of $\beta$ and allows consistent comparison across baselines.

## Acknowledgements

We would like to thank the anonymous reviewers for their thoughtful comments on the draft version of this paper. We would also like to thank Daniel Mankowitz for suggesting relevant research in the area of robust and risk-aware reinforcement learning and for providing insightful comments during the development of the paper.

## Funding Transparency Statement

Michael Gimelfarb was funded by an Ontario Graduate Scholarship, a DiDi Graduate Student Award and a Vector Institute Postgraduate Affiliate Award, Scott Sanner was funded by an Ontario Early Researcher Award Grant, and Chi-Guhn Lee was funded by an NSERC Discovery Grant.

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
