# Risk-Aware Transfer in Reinforcement Learning using Successor Features

## Supplementary Material

## Abstract

This part of the paper discusses algorithmic details for the total reward and the discounted setting that were not included in the main paper due to space limitations. It includes proofs of all main theoretical claims, as well as all domain configurations and parameter settings that are required to reproduce the experiments. Finally, it includes additional experiments and ablation studies that had to be excluded from the main paper due to space limitations, and the NeurIPS paper checklist.

## Contents

# A  Mathematical and Algorithmic Details

In this section, we outline the ways in which successor features and their covariance matrices can be learned in practical RL settings. Specifically, we discuss the details surrounding Bellman updates and the distributional RL framework, and also discuss the mean-variance approximation under more general assumptions in parameteric density estimation.

## A.1  Mean-Variance Approximation for Episodic MDPs

**Directly Computing Successor Features and Covariances.**  In the tabular setting, we compute estimates of the mean $\tilde{\psi}_h^\pi(s, a)$ and covariance $\tilde{\Sigma}_h^\pi(s, a)$ by extending the analysis in Sherstan et al. [37] to the $d$-dimensional setting. For a transition $(s, a, \phi, s', a')$, where $\phi = \phi(s, a, s')$, and learning rate $\alpha > 0$, the update for successor features is:

$$
\begin{aligned}
\tilde{\boldsymbol{\delta}}_h &= \boldsymbol{\phi} + \tilde{\psi}_{h+1}^\pi(s', a') - \tilde{\psi}_h^\pi(s, a), \\
\tilde{\psi}_h^\pi(s, a) &= \tilde{\psi}_h^\pi(s, a) + \alpha\tilde{\boldsymbol{\delta}}_h.
\end{aligned}
\tag{9}
$$

By virtue of the covariance Bellman equation (8), a per-sample update of the covariance matrix can be computed by using the Bellman residuals $\tilde{\boldsymbol{\delta}}_h$ as a pseudo-reward:

$$
\begin{aligned}
\Delta_h &= \tilde{\boldsymbol{\delta}}_h\tilde{\boldsymbol{\delta}}_h^\mathsf{T} + \tilde{\Sigma}_{h+1}^\pi(s', a') - \tilde{\Sigma}_h^\pi(s, a), \\
\tilde{\Sigma}_h^\pi(s, a) &= \tilde{\Sigma}_h^\pi(s, a) + \bar{\alpha}\Delta_h.
\end{aligned}
\tag{10}
$$

In practice, $\bar{\alpha}$ is usually set much smaller than $\alpha$, e.g. $\bar{\alpha} = \rho\alpha$ for some positive $\rho \ll 1$.

**Computing Reward Parameters.**  When the reward parameters $\mathbf{w}$ are unknown, they can be learned by solving a regression problem. Specifically, for known or estimated features $\phi(s, a, s')$ and an observed reward $r$, the objective function to minimize is the *mean-squared error*,

$$
\mathcal{L}(\mathbf{w}) = \frac{1}{2}\left(r - \phi(s, a, s')^\mathsf{T}\mathbf{w}\right)^2,
$$

that can be minimized using *stochastic gradient descent* (SGD). Introducing a learning rate $\alpha_w > 0$, an update of SGD on a single transition $(s, a) \rightarrow s'$ is

$$
\mathbf{w} \leftarrow \mathbf{w} + \alpha_w(r - \phi(s, a, s')^\mathsf{T}\mathbf{w})\phi(s, a, s').
$$

**Pseudocode.**  The general routine for performing risk-aware transfer learning in an online setting, which we call *Risk-Aware Successor Feature Q-Learning* (RaSFQL), can now be fully described. Pseudocode adapted for the total-reward episodic MDP setting is given as Algorithm 1. Please note that our approach closely follows the risk-neutral SFQL in Barreto et al. [2].

Both the discounted and total reward episodic settings are amenable to function approximation. However, as discussed in the main text, this "residual" method is usually not advisable as the approximation errors in the residuals $\tilde{\boldsymbol{\delta}}_h$ can dominate the environment uncertainty. While this could be useful for handling *epistemic* or model uncertainty in successor features [21], the intent of our work is to learn the *aleatory* or environment uncertainty. Therefore, a more precise method — based on the projected Bellman equation — will be introduced in Appendix A.3.

## A.2  Mean-Variance Approximation for Discounted MDPs

**Bellman Principle and Augmented MDP.**  The utility objective in the discounted infinite-horizon setting becomes

$$
\mathcal{Q}_{\beta,\gamma}^\pi(s, a) = U_\beta\left[\sum_{t=0}^\infty \gamma^t r(s_t, a_t, s_{t+1})\right],
$$

where $\gamma \in (0, 1)$ is a discount factor for future rewards.

In the discounted setting, it is necessary to accumulate and keep track of the discounting over time. This can be implemented by augmenting the state space of the original MDP [5]. Specifically, define

**Algorithm 1** RaSFQL with Mean-Variance Approximation

---

1: **Requires** $m, T, N_e \in \mathbb{N}$, $\varepsilon \in [0,1]$, $\alpha, \bar{\alpha}, \alpha_w > 0$, $\beta \in \mathbb{R}$, $\phi \in \mathbb{R}^d$, $M_1, \ldots M_m \in \mathcal{M}$
2: **for** $t = 1, 2 \ldots m$ **do**
   \\ Initialize successor features and covariance for the current task
3:     **if** $t = 1$ **then** Initialize $\tilde{\psi}^t, \tilde{\Sigma}^t$ to small random values **else** Initialize $\tilde{\psi}^t, \tilde{\Sigma}^t$ to $\tilde{\psi}^{t-1}, \tilde{\Sigma}^{t-1}$
4:     Initialize $\tilde{\mathbf{w}}_t$ to small random values
   \\ Commence training on task $M_t$
5:     **for** $n_e = 1, 2 \ldots N_e$ **do**
6:         Initialize task $M_t$ with initial state $s$
7:         **for** $h = 0, 1 \ldots T$ **do**
      \\ Select the source task $M_c$ using GPI
8:             $c \leftarrow \arg\max_j \max_b \{ \tilde{\psi}_h^j(s,b)^\mathsf{T} \tilde{\mathbf{w}}_t - \beta \tilde{\mathbf{w}}_t^\mathsf{T} \tilde{\Sigma}_h^j(s,b) \tilde{\mathbf{w}}_t \}$
      \\ Sample action from the epsilon-greedy policy based on $\pi_c$
9:             random_a $\sim \mathrm{Bernoulli}(\varepsilon)$
10:            **if** random_a **then** $a \sim \mathrm{Uniform}(\mathcal{A})$ **else**
                $a \leftarrow \arg\max_b \{ \tilde{\psi}_h^c(s,b)^\mathsf{T} \tilde{\mathbf{w}}_t - \beta \tilde{\mathbf{w}}_t^\mathsf{T} \tilde{\Sigma}_h^c(s,b) \tilde{\mathbf{w}}_t \}$
11:            Take action $a$ in $M_t$ and observe $r$ and $s'$
    \\ Update reward parameters for the current task
12:            $\tilde{\mathbf{w}}_t \leftarrow \tilde{\mathbf{w}}_t + \alpha_w (r - \phi(s,a,s')^\mathsf{T} \tilde{\mathbf{w}}_t) \phi(s,a,s')$
    \\ Update the successor features and covariance for the current task
13:            $a' \leftarrow \arg\max_b \max_j \{ \tilde{\psi}_h^j(s',b)^\mathsf{T} \tilde{\mathbf{w}}_t - \beta \tilde{\mathbf{w}}_t^\mathsf{T} \tilde{\Sigma}_h^j(s',b) \tilde{\mathbf{w}}_t \}$
14:            Update $\tilde{\psi}_h^t, \tilde{\Sigma}_h^t$ on $(s, a, \phi, s', a')$ using (9) and (10)
    \\ Update the successor features and covariance for task $M_c$
15:            **if** $c \neq t$ **then**
16:                $a' \leftarrow \arg\max_b \{ \tilde{\psi}_h^c(s',b)^\mathsf{T} \tilde{\mathbf{w}}_c - \beta \tilde{\mathbf{w}}_c^\mathsf{T} \tilde{\Sigma}_h^c(s',b) \tilde{\mathbf{w}}_c \}$
17:                Update $\tilde{\psi}_h^c, \tilde{\Sigma}_h^c$ on $(s, a, \phi, s', a')$ using (9) and (10)
18:            **end if**
19:            $s \leftarrow s'$
20:         **end for**
21:     **end for**
22: **end for**

---

$\mathcal{Z} = [0, 1]$ and let $z \in \mathcal{Z}$ denote the state of discounting. For a given MDP $\langle \mathcal{S}, \mathcal{A}, r, P \rangle$, we define the augmented MDP $\langle \mathcal{S}', \mathcal{A}, r', P' \rangle$, with state space $\mathcal{S}' = \mathcal{S} \times \mathcal{Z}$, action space $\mathcal{A}$, reward function

$$r'((s,z), a, (s',z')) = zr(s,a,s'),$$

and dynamics

$$P'((s',z')|(s,z),a) = P(s'|s,a)\delta_{\gamma z}(z'),$$

where $\delta$ is the Dirac delta function. Applying this augmentation transformation to a set of MDPs with common transition function and common discount factor implies that the set of augmented MDPs will also have the same transition functions.

Moreover, the following Bellman equation can be derived for the augmented MDP [5]:

$$
\begin{aligned}
\mathcal{J}_\beta^\pi(s,a,z) &= U_\beta \left[ zr(s,a,s') + \mathcal{J}_\beta^\pi(s', \pi(s', \gamma z), \gamma z) \right] \\
&= \frac{1}{\beta} \log \mathbb{E}_{s' \sim P(\cdot|s,a)} \left[ \exp\left\{ \beta \left( zr(s,a,s') + \mathcal{J}_\beta^\pi(s', \pi(s', \gamma z), \gamma z) \right) \right\} \right].
\end{aligned}
\tag{11}
$$

Then, we can recover the original utility with $\mathcal{Q}_{\beta,\gamma}^\pi(s,a) = \mathcal{J}_\beta^\pi(s,a,1)$. Furthermore, the Bellman equation above converges to a unique fixed point, and so the search for optimal policies can be restricted to stationary Markov policies $\pi : \mathcal{S} \times \mathcal{Z} \to \mathcal{A}$.

However, learning general policies $\pi : \mathcal{S} \times \mathcal{Z} \to \mathcal{A}$ introduces additional difficulties in the function approximation setting. In this case, successor features and their covariance matrices would have to be functions of $z$. For a single transition, their corresponding updates would also require a sweep over all possible values of $z$, e.g. $z = 1, \gamma, \gamma^2, \ldots$, and would be computationally demanding. On the other hand, restricting the search to stationary policies $\pi : \mathcal{S} \to \mathcal{A}$ alleviates this computational

burden, making the overall time and space complexity per update comparable to the risk-neutral SF representation, and also allows off-the-shelf RL algorithms to be used to learn successor features. This also facilitates more precise estimation of risk using the distributional framework discussed in the next section.

Fortunately, the restriction to $z$-independent source policies does not affect the validity of Theorem 1, since policy improvement was shown for *arbitrary* admissible policies. This implies that monotone policy improvement is guaranteed even for $z$-independent policies, provided that their utilities can be estimated. In the case of Theorem 2, the approximation error $\varepsilon$ generally arises from two sources of additive error, namely that of restricting optimal policies $\pi_i^*$ to $z$-independent optimal policies $\bar{\pi}_i^*$, and that of approximating utilities using function approximation, e.g.

$$
\begin{aligned}
\varepsilon &= \left| \tilde{\mathcal{J}}_\beta^{\bar{\pi}^*_i}(s, a, 1) - \mathcal{J}_\beta^{\pi^*_i}(s, a, 1) \right| \\
&= \left| \tilde{\mathcal{J}}_\beta^{\bar{\pi}^*_i}(s, a, 1) - \mathcal{J}_\beta^{\bar{\pi}^*_i}(s, a, 1) + \mathcal{J}_\beta^{\bar{\pi}^*_i}(s, a, 1) - \mathcal{J}_\beta^{\pi^*_i}(s, a, 1) \right| \\
&\leq \left| \tilde{\mathcal{J}}_\beta^{\bar{\pi}^*_i}(s, a, 1) - \mathcal{J}_\beta^{\bar{\pi}^*_i}(s, a, 1) \right| + \left| \mathcal{J}_\beta^{\bar{\pi}^*_i}(s, a, 1) - \mathcal{J}_\beta^{\pi^*_i}(s, a, 1) \right| \\
&= \left\{ \text{approximation error of } \mathcal{J}_\beta^{\bar{\pi}_i} \right\} + \left\{ \text{absolute difference between utilities of } \bar{\pi}_i^* \text{ and } \pi_i^* \right\}.
\end{aligned}
$$

The first source of error arises solely due to the method of function approximation, and can be reduced by using architectures whose training parameters and capacity are well-calibrated for each problem. The second source of error is in general irreducible, but whether it can be tolerated should be traded-off against the difficulty of learning $z$-dependent policies. In general, the learning of $z$-dependent policies tractably is a challenging problem, which we leave for future investigation.

**Incorporating Moment Information into GPE in Discounted MDPs.** We now apply the idea of generalized policy evaluation to discounted objectives. First, observe that for fixed $\pi : \mathcal{S} \to \mathcal{A}$:

$$
\mathcal{J}_\beta^\pi(s, a, 1) = U_\beta \left[ \sum_{t=0}^\infty \gamma^t r(s_t, \pi(s_t), s_{t+1}) \right] = U_\beta \left[ \Psi^\pi(s, a)^\intercal \mathbf{w} \right], \tag{12}
$$

corresponding to the random vector $\Psi^\pi(s, a) = \sum_{t=0}^\infty \gamma^t \phi_t$ of unrealized feature returns at time $h$. Thus, we have again transformed the problem of estimating the utility of rewards into the problem of estimating the moments of the random variable $\Psi^\pi(s, a)^\intercal \mathbf{w}$.

Next, computing the Taylor expansion of $U_\beta$:

$$
\begin{aligned}
\mathcal{J}_\beta^\pi(s, a, 1) &= \mathbb{E}_P[\Psi^\pi(s, a)^\intercal \mathbf{w}] + \frac{\beta}{2} \text{Var}_P[\Psi^\pi(s, a)^\intercal \mathbf{w}] + O(\beta^2) \\
&\approx \boldsymbol{\psi}^\pi(s, a)^\intercal \mathbf{w} + \frac{\beta}{2} \mathbf{w}^\intercal \text{Var}_P[\Psi^\pi(s, a)] \mathbf{w} = \tilde{\mathcal{J}}_\beta^\pi(s, a, 1). 
\end{aligned} \tag{13}
$$

From a practical point of view, the mean-variance approximation in the discounted setting is identical to the episodic total-reward setting, with the exception that the successor features and covariance are discounted (and also time-independent). As in the undiscounted case, (13) induces an error of $O(\beta^2)$, but is now another instantiation of GPE. However, restricting the search to $z$-independent policies introduces additional approximation error that can also be absorbed into $\varepsilon$, as discussed previously. Crucially, the theoretical results proved for the discounted setting (Appendix B.3) will now also hold for $z$-independent stationary policies.

**Bellman Updates for Covariance in Discounted MDPs.** The covariance matrix satisfies the covariance Bellman equation

$$
\Sigma_h^\pi(s, a) = \mathbb{E}_{s' \sim P(\cdot|s,a)} \left[ \boldsymbol{\delta}_h \boldsymbol{\delta}_h^\intercal + \gamma^2 \Sigma_{h+1}^\pi(s', \pi_{h+1}(s')) \mid s_h = s, a_h = a \right], \tag{14}
$$

Similar to (9) and (10), in the discounted setting the successor features can be computed as [2]:

$$
\begin{aligned}
\tilde{\boldsymbol{\delta}}_h &= \boldsymbol{\phi} + \gamma \tilde{\boldsymbol{\psi}}_{h+1}^\pi(s', a') - \tilde{\boldsymbol{\psi}}_h^\pi(s, a), \\
\tilde{\boldsymbol{\psi}}_h^\pi(s, a) &= \tilde{\boldsymbol{\psi}}_h^\pi(s, a) + \alpha \tilde{\boldsymbol{\delta}}_h.
\end{aligned} \tag{15}
$$

Once again, the covariance matrix can be updated per sample following (14):

$$\Delta_h = \tilde{\boldsymbol{\delta}}_h \tilde{\boldsymbol{\delta}}_h^{\mathsf{T}} + \gamma^2 \tilde{\Sigma}_{h+1}^{\pi}(s', a') - \tilde{\Sigma}_h^{\pi}(s, a),$$
$$\tilde{\Sigma}_h^{\pi}(s, a) = \tilde{\Sigma}_h^{\pi}(s, a) + \bar{\alpha}\Delta_h. \tag{16}$$

In the context of Algorithm 1, all calls to (9) and (10) would be replaced with (15) and (16), respectively.

The convergence of the covariance matrix in the discounted setting (14) is established in the following result that can be easily proved using the techniques in Appendix B.4 for the episodic setting.

**Theorem 4** (**Convergence of Covariance**). *Let* $\| \cdot \|$ *be a matrix-compatible norm, and suppose there exists* $\varepsilon : \mathcal{S} \times \mathcal{A} \times \mathcal{T} \to [0, \infty)$ *such that*

1. $\|\tilde{\boldsymbol{\psi}}_h^{\pi}(s, a) - \boldsymbol{\psi}_h^{\pi}(s, a)\|^2 \le \varepsilon_h(s, a)$

2. $\|\mathbb{E}_{s' \sim P(\cdot|s,a)}[\gamma \tilde{\boldsymbol{\delta}}_h(\tilde{\boldsymbol{\psi}}_h^{\pi}(s', \pi_{h+1}(s')) - \boldsymbol{\psi}_h^{\pi}(s', \pi_{h+1}(s')))^{\mathsf{T}}]\| \le \varepsilon_h(s, a).$

*Then,*

$$\left\| \Sigma_h^{\pi}(s, a) - \mathbb{E}_{s' \sim P(\cdot|s,a)} \left[ \tilde{\boldsymbol{\delta}}_h \tilde{\boldsymbol{\delta}}_h^{\mathsf{T}} + \gamma^2 \tilde{\Sigma}_{h+1}^{\pi}(s', \pi_{h+1}(s')) \right] \right\| \le 3\varepsilon_h(s, a).$$

Please note that this result is identical to Theorem 3, with the exception of the discount factor.

## A.3 Histogram Representations for Successor Features

The theoretical framework for distributional RL is discussed in details in the relevant literature [6]. In this appendix, we discuss how this framework can be applied to learn distributions over successor features, and how to use these distributions to select actions in a risk-aware manner.

**Learning Distributions over Successor Features.** As discussed in the main text, the goal is to estimate the distribution of each component in the discounted infinite horizon setting

$$\Psi_i^{\pi}(s, a) = \sum_{t=0}^{\infty} \gamma^t \phi_i(s_t, a_t, s_{t+1}),$$

starting from $s_0 = s$, $a_0 = a$, where $a_t = \pi(s_t)$ is selected according to a policy $\pi$. Treating $\Psi_1^{\pi}, \ldots \Psi_d^{\pi}$ as value functions, we are now able to apply distributional RL.

Specifically, suppose that each state feature component is bounded in a compact interval, e.g. $\phi_i(s, a, s') \in [\phi_i^{min}, \phi_i^{max}]$. Then, we may define corresponding bounds on $\Psi_i^{\pi}(s, a)$ by bounding the terms of its geometric series representation above:

$$\Psi_i^{min} = \frac{\phi_i^{min}}{1 - \gamma} \le \Psi_i^{\pi}(s, a) \le \frac{\phi_i^{max}}{1 - \gamma} = \Psi_i^{max}.$$

Now, we may model each $\Psi_h^{\pi}(s, a)$ by using a discrete distribution parameterized by $N \in \mathbb{N}$ and $[\Psi_i^{min}, \Psi_i^{max}]$, whose support is defined by a set of atoms

$$\mathcal{Z}_i = \left\{ z_{j,i} = \Psi_i^{min} + j\Delta z_i : 0 \le j < N \right\}, \Delta z_i = \frac{\Psi_i^{max} - \Psi_i^{min}}{N - 1}, \forall i = 1, \ldots d.$$

Finally, the atom probabilities for $z_{j,i}$ are given by a parameteric model $\theta_{j,i} : \mathcal{S} \times \mathcal{A} \to \mathbb{R}^N$, e.g.

$$Z_{\theta,i}(s, a) = z_{j,i} \quad \text{w.p. } p_{j,i}(s, a) = \frac{e^{\theta_{j,i}(s,a)}}{\sum_j e^{\theta_{j,i}(s,a)}}, \tag{17}$$

where the softmax layer ensures that probabilities are non-negative and sum to one.

In order to update $p_{j,i}$ on environment transitions $(s, a, \phi_i, s')$, we project the Bellman updates for each $i$ onto the support of $\mathcal{Z}_i$. To do this, given a sample $(s, a, \phi_i, s')$, we compute the projected Bellman update, clipped to the interval $[\Psi_i^{min}, \Psi_i^{max}]$

$$\hat{\mathcal{T}}_i z_{j,i} = \text{clip} \left( \phi_i + \gamma z_{j,i}; [\Psi_i^{min}, \Psi_i^{max}] \right),$$

and then distribute its probability $p_{j,i}(s', \pi(s'))$ to the immediate neighbors of $\hat{\mathcal{T}}_i z_{j,i}$. Here, we again follow Bellemare et al. [6] and define the projected operator $\Phi$ with $j$-th component equal to

$$(\Phi \hat{\mathcal{T}}_i Z_{\theta,i}(s,a))_j = \sum_{k=0}^{N-1} \text{clip}\left(1 - \frac{\left|\hat{\mathcal{T}}_i z_{k,i} - z_{j,i}\right|}{\Delta z_i}; [0,1]\right) p_{k,i}(s', \pi(s')).$$

As standard in deep RL, we view the target distribution $p_{k,i}(s', \pi(s'))$ as parameterized by a set of frozen parameters $\theta'$. Then, the loss function to optimize for the sample $(s, a, \phi_i, s')$ is given as the cross-entropy term

$$\mathcal{L}_i(\theta) = D_{KL}\left(\Phi \hat{\mathcal{T}}_i Z_{\theta',i}(s,a) \, \Big\| \, Z_{\theta,i}(s,a)\right),$$

that can be easily optimized using gradient descent.

**Calculating Utilities.** The calculation of (7) is a trivial matter given the distribution (17). In particular, we have:

$$\mathbb{E}[Z_{\theta,i}(s,a)^p] = \sum_{j=0}^{N-1} (z_{j,i})^p p_{j,i}(s,a), \quad p \in \mathbb{N}, \tag{18}$$

from which we can easily compute the variance

$$\text{Var}[\Psi_i^\pi(s,a)] = \text{Var}[Z_{\theta,i}(s,a)] = \text{E}[Z_{\theta,i}(s,a)^2] - \text{E}[Z_{\theta,i}(s,a)]^2. \tag{19}$$

Recall that by the independence assumption, the cross-covariance terms are ignored in these calculations, and thus $\Sigma_i^\pi(s,a)$ is represented as a diagonal matrix with entries on the $i$-th diagonal term equal to $\text{Var}[\Psi_i^\pi(s,a)]$.

Another possibility is to compute the entropic utility $U_\beta$ exactly. In particular, using the independence assumption of $\Psi_i^\pi(s,a)$ again, we have:

$$U_\beta[\Psi^\pi(s,a)^\intercal \mathbf{w}] = \frac{1}{\beta} \log \mathbb{E}\left[e^{\beta \Psi^\pi(s,a)^\intercal \mathbf{w}}\right] \approx \sum_{i=1}^d \frac{1}{\beta} \log \mathbb{E}\left[e^{\beta \Psi_i^\pi(s,a) w_i}\right]$$

$$= \sum_{i=1,\, w_i \neq 0}^d w_i \frac{1}{\beta w_i} \log \mathbb{E}\left[e^{(\beta w_i)\Psi_i^\pi(s,a)}\right] = \sum_{i=1}^d w_i U_{\beta w_i}[\Psi_i^\pi(s,a)], \tag{20}$$

and can be seen as another risk-sensitive instantiation of GPE. Crucially, the utility terms in (20) can be calculated efficiently in the C51 framework using (17)

$$U_\beta[\Psi_i^\pi(s,a)] = \frac{1}{\beta} \log \mathbb{E}\left[e^{\beta Z_{\theta,i}(s,a)}\right] = \frac{1}{\beta} \log \sum_{j=0}^{N-1} e^{\beta z_{j,i}} p_{j,i}(s,a).$$

However, this quantity is difficult to compute numerically, since for negative $\beta$, the terms $e^{\beta z_{j,i}}$ often suffer from overflow at $z_{j,i}$ close to $\Psi_i^{min}$, and underflow for $z_{j,i}$ close to $\Psi_i^{max}$. This becomes considerably more problematic for $\beta$ of larger magnitude, such as when risk-awareness is a priority, or for rewards $\mathbf{w}$ of larger magnitude. We also find that the log-sum-exp trick, a standard computational device used for calculations of this form, offers relatively little improvement. A similar issue has also been previously pointed out in other work using the entropic utility [52]. For this reason, we use the mean-variance approximation, which provides an excellent approximation to the entropic utility for various values of $\beta$, as we demonstrated experimentally, and without suffering from the aforementioned issues above.

**Pseudocode.** The approach described above can be applied to compute the distribution of successor features for every component $i = 1, \ldots d$ across all training task instances. This results in a new algorithm that we call SFC51. Generally, the training procedure of SFC51 is identical in structure to SFDQN in Barreto et al. [2], except the deterministic DQN update of successor features [55] is replaced by the distributional C51 update described above. Therefore, the overall training procedure is similar to Algorithm 1, but with a few subtle differences. First, instead of learning $\mathbf{w}$, it is provided to the agent as done in SFDQN. Second, every sample $(s, a, \phi, s')$ collected from any training is

**Algorithm 2** RaSFC51 with Mean-Variance Approximation

---

1: **Requires** $m, T, N, N_e \in \mathbb{N}$, $\varepsilon \in [0,1]$, $\beta, \phi_1^{min}, \phi_1^{max}, \ldots \phi_d^{min}, \phi_d^{max} \in \mathbb{R}$, $\boldsymbol{\phi} \in \mathbb{R}^d$, $\gamma \in (0,1)$, $M_1, \ldots M_m \in \mathcal{M}$ with $\mathbf{w}_1, \ldots \mathbf{w}_m \in \mathbb{R}^d$
$\quad$ \\ Initialize atoms and their probability distributions
2: Initialize $\boldsymbol{\theta}^1(s,a), \ldots \boldsymbol{\theta}^m(s,a)$ to random values
3: **for** $i = 1, 2 \ldots d$ **do** $\Psi_i^{min} \leftarrow \frac{\phi_i^{min}}{1-\gamma}$, $\Psi_i^{max} \leftarrow \frac{\phi_i^{max}}{1-\gamma}$, $\Delta z_i \leftarrow \frac{\Psi_i^{max} - \Psi_i^{min}}{N-1}$
4: **for** $i = 1, 2 \ldots d$ **do for** $j = 0, 1 \ldots N-1$ **do** $z_{j,i} \leftarrow \Psi_i^{min} + j\Delta z_i$
$\quad$ \\ Main training loop
5: **for** $t = 1, 2 \ldots m$ **do**
$\quad$ \\ Commence training on task $M_t$
6: $\quad$ **for** $n_e = 1, 2 \ldots N_e$ **do**
7: $\quad\quad$ Initialize task $M_t$ with initial state $s$
8: $\quad\quad$ **for** $h = 0, 1 \ldots T$ **do**
$\quad\quad\quad$ \\ Extract sufficient statistics from $\boldsymbol{\theta}^t(s, \cdot)$ and select the source task $M_c$ using GPI
9: $\quad\quad\quad$ **for** $j = 1, 2 \ldots m$ **do** Compute $\tilde{\boldsymbol{\psi}}^j(s, \cdot)$, $\tilde{\Sigma}^j(s, \cdot)$ using $\boldsymbol{\theta}^j(s, \cdot)$ and (18) and (19)
10: $\quad\quad\quad$ $c \leftarrow \arg\max_j \max_b \{\tilde{\boldsymbol{\psi}}^j(s,b)^\mathsf{T} \mathbf{w}_t - \beta \mathbf{w}_t^\mathsf{T} \tilde{\Sigma}^j(s,b) \mathbf{w}_t\}$
$\quad\quad\quad$ \\ Sample action from the epsilon-greedy policy based on $\pi_c$
11: $\quad\quad\quad$ random_a $\sim$ Bernoulli($\varepsilon$)
12: $\quad\quad\quad$ **if** random_a **then** $a \sim$ Uniform($\mathcal{A}$) **else**
$\quad\quad\quad$ $a \leftarrow \arg\max_b \{\tilde{\boldsymbol{\psi}}^c(s,b)^\mathsf{T} \mathbf{w}_t - \beta \mathbf{w}_t^\mathsf{T} \tilde{\Sigma}^c(s,b) \mathbf{w}_t\}$
13: $\quad\quad\quad$ Take action $a$ in $M_t$ and observe $r$ and $s'$
$\quad\quad\quad$ \\ Update $\boldsymbol{\theta}^1(s,a), \ldots \boldsymbol{\theta}^m(s,a)$
14: $\quad\quad\quad$ **for** $c = 1, 2 \ldots m$ **do**
$\quad\quad\quad\quad$ \\ Extract sufficient statistics from $\boldsymbol{\theta}^c(s', \cdot)$ and select action $a' = \pi_c(s')$
15: $\quad\quad\quad\quad$ Compute $\tilde{\boldsymbol{\psi}}^c(s', \cdot)$, $\tilde{\Sigma}^c(s', \cdot)$ using $\boldsymbol{\theta}^c(s', \cdot)$ and (18) and (19)
16: $\quad\quad\quad\quad$ $a' \leftarrow \arg\max_b \{\tilde{\boldsymbol{\psi}}^c(s',b)^\mathsf{T} \mathbf{w}_c - \beta \mathbf{w}_c^\mathsf{T} \tilde{\Sigma}^c(s',b) \mathbf{w}_c\}$
$\quad\quad\quad\quad$ \\ Apply Categorical Algorithm to update $\boldsymbol{\theta}^c(s,a)$
17: $\quad\quad\quad\quad$ **for** $j = 0, 1 \ldots N-1$ **do for** $i = 1, \ldots d$ **do** $m_{j,i} \leftarrow 0$
18: $\quad\quad\quad\quad$ **for** $i = 1, 2 \ldots d$ **do**
19: $\quad\quad\quad\quad\quad$ **for** $j = 0, 1 \ldots N-1$ **do**
$\quad\quad\quad\quad\quad\quad$ \\ Compute the projection of $\hat{\mathcal{T}}_i z_{j,i}$ onto the support $\mathcal{Z}_i$
20: $\quad\quad\quad\quad\quad\quad$ $\hat{\mathcal{T}}_i z_{j,i} \leftarrow \text{clip}\left(\phi_i(s,a,s') + \gamma z_{j,i}; [\Psi_i^{min}, \Psi_i^{max}]\right)$
21: $\quad\quad\quad\quad\quad\quad$ $b_{j,i} \leftarrow (\hat{\mathcal{T}}_i z_{j,i} - \Psi_i^{min})/\Delta z_i$
22: $\quad\quad\quad\quad\quad\quad$ $l \leftarrow \lfloor b_{j,i} \rfloor$, $u \leftarrow \lceil b_{j,i} \rceil$
$\quad\quad\quad\quad\quad\quad$ \\ Distribute probability of $\hat{\mathcal{T}}_i z_{j,i}$
23: $\quad\quad\quad\quad\quad\quad$ $m_{l,i} \leftarrow m_{l,i} + p_{j,i}^c(s', a')(u - b_{j,i})$
24: $\quad\quad\quad\quad\quad\quad$ $m_{u,i} \leftarrow m_{u,i} + p_{j,i}^c(s', a')(b_{j,i} - l)$
25: $\quad\quad\quad\quad\quad$ **end for**
26: $\quad\quad\quad\quad$ **end for**
27: $\quad\quad\quad\quad$ Backpropagate through $-\sum_{j,i} m_{j,i} \log p_{j,i}^c(s,a)$ to update $\boldsymbol{\theta}^c(s,a)$
28: $\quad\quad\quad$ **end for**
29: $\quad\quad\quad$ $s \leftarrow s'$
30: $\quad\quad$ **end for**
31: $\quad$ **end for**
32: **end for**

---

used to update all successor feature distributions simultaneously, as also done in SFDQN. Finally, the utility of returns can be used to select actions, rather than the expected return as done in DQN. Applying this last modification to SFC51 leads our proposed algorithm, which we call *Risk-aware SFC51* (RaSFC51). Of course, SFC51 can be recovered by simply setting $\beta = 0$. A complete description of RaSFC51 with the mean-variance approximation is provided in Algorithm 2[3].

---

[3]In practice, the double for loop starting in lines 18 and 19 can be implemented efficiently by vectoring the computation of $m_{j,i}$, in languages that support vectorized arithmetic operations.

## A.4 Possible Generalizations of the Mean-Variance Approximation

**Cumulant-Generating Functions.** The quantity $K_R(\beta) = \log \mathbb{E}[e^{\beta R}]$ in (1) is often referred to as the *cumulant-generating function*. The cumulant generating function admits the well-known Taylor expansion:

$$U_\beta[R] = \frac{1}{\beta} K_R(\beta) = \frac{1}{\beta} \sum_{n=1}^{\infty} \kappa_R(n) \frac{\beta^n}{n!} = \sum_{n=1}^{\infty} \kappa_R(n) \frac{\beta^{n-1}}{n!}, \tag{21}$$

where $\kappa_R(n)$ is the $n$-th *cumulant* of the random variable $R$ [62]. The mean-variance approximation (7) then follows directly from (21) by ignoring all terms of order $n \geq 3$. Another way to look at the mean-variance approximation is that it is the result of applying a *Laplace approximation* to the return distribution prior to calculating its utility [48]. While it is also possible to approximate (21) using orders of $n$ greater than 2, such approximations would no longer provide "instantaneous" GPE. In particular, cumulants are much harder to compute as functions of $\mathbf{w}$ for $n = 3$ [49], and no closed formulas are even known to us for $n \geq 4$.

**Elliptical Distributions.** The mean-variance approximation (7) results from making the distributional assumption $\Psi_h^\pi(s,a) \sim \mathcal{N}(\boldsymbol{\psi}_h^\pi(s,a), \Sigma_h^\pi(s,a))$. Since the normal distribution is a member of the class of elliptical distributions, a natural question to ask is whether GPE can apply to other members of this class of distributions as well.

Formally, a random variable $X$ has an *elliptical distribution* on $\mathbb{R}^d$ if there exists $\boldsymbol{\mu} \in \mathbb{R}^d$, positive definite $\Sigma \in \mathbb{R}^{d \times d}$ and a positive-valued function $\xi : \mathbb{R} \to \mathbb{R}$, and the characteristic function of $X$ has the form

$$\mathbb{E}[e^{i\mathbf{t}^\mathsf{T} X}] = e^{i\mathbf{t}^\mathsf{T}\boldsymbol{\mu}}\xi(\mathbf{t}^\mathsf{T}\Sigma\mathbf{t}), \quad \forall \mathbf{t} \in \mathbb{R}^d. \tag{22}$$

Equivalently, for any random variable with characteristic function (22), there exists a positive function $g_d : \mathbb{R} \to \mathbb{R}$ such that the density of $X$ is

$$f_X(\mathbf{x}) \propto |\Sigma|^{-1/2} g_d\left((\mathbf{x} - \boldsymbol{\mu})^\mathsf{T} \Sigma^{-1} (\mathbf{x} - \boldsymbol{\mu})\right).$$

In either case, we write $X \sim \mathcal{E}_d(\boldsymbol{\mu}, \Sigma, \xi)$. One advantage of this parameterization is that $\boldsymbol{\mu}$ corresponds exactly to the mean of $X$, e.g. $\mathbb{E}[X] = \boldsymbol{\mu}$. Furthermore, if the covariance of $X$ exists, then it is equal to $\Sigma$ up to a positive multiplicative constant, e.g. $\mathrm{Var}[X] = c\Sigma$ for some $c > 0^4$.

In order to connect this to the SF framework, we parameterize $\Psi_h^\pi(s,a) \sim \mathcal{E}_d(\boldsymbol{\psi}_h^\pi(s,a), \Sigma_h^\pi(s,a), \xi)$. Then, using the linearity property (6, 12), GPE evaluates the entropic utilities of the random variables $\Psi_h^\pi(s,a)^\mathsf{T}\mathbf{w}$. Fortunately, affine transforms of elliptically distributed random variables are univariate elliptically distributed [57].

**Lemma 2.** *Let $X \sim \mathcal{E}_d(\boldsymbol{\mu}, \Sigma, \xi)$ and $\mathbf{w} \in \mathbb{R}^d$. Then, $X^\mathsf{T}\mathbf{w} \sim \mathcal{E}_1(\boldsymbol{\mu}^\mathsf{T}\mathbf{w}, \mathbf{w}^\mathsf{T}\Sigma\mathbf{w}, \xi)$.*

Applying Lemma 2 and then substituting $t = -i\beta$, the entropic utility becomes

$$U_\beta[\Psi_h^\pi(s,a)^\mathsf{T}\mathbf{w}] = \boldsymbol{\psi}_h^\pi(s,a)^\mathsf{T}\mathbf{w} + \frac{1}{\beta}\log\xi\left(-\beta^2 \mathbf{w}^\mathsf{T}\Sigma_h^\pi(s,a)\mathbf{w}\right), \tag{23}$$

and is also a mean-variance approximation. However, unlike (7) in which $\frac{1}{\beta}\log \circ \xi$ is the identity mapping, (23) is allowed to depend *non-linearly* on the variance of returns. Table 2 illustrates these mappings for different distributional assumptions. For heavy-tailed distributions, $\xi$ should increase super-linearly for sufficiently large return variances, and thus (23) will often be more sensitive to variance than (7). This phenomenon is clearly illustrated in Figure 8 by comparing the variance penalties of the normal and Laplace distributions. Another advantage of this generalization is that the methodologies for estimating successor features and their covariances (Appendix A.1 and A.3) can be directly applied to this more general setting. An ablation study comparing the Gaussian assumption and the Laplace assumption in a complex experiment is provided at the end of Appendix D.2.

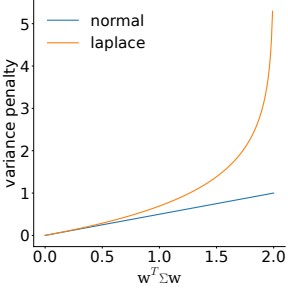

Figure 8: Comparing $\frac{1}{\beta}\log \circ \xi$ in (23) for normal and Laplace distributions, for $\beta = 1$.

---

[4]This implies that the Bellman updates (10) or (16) can still be used to learn $\Sigma_h^\pi$, but now the resulting estimates must be scaled by $c$ when computing the utilities, if $c$ is not one.

| Name | Parameters | $\frac{1}{\beta} \log \circ \xi$ |
|---|---|---|
| multivariate normal | $\boldsymbol{\mu}, \boldsymbol{\Sigma}$ | $\frac{\beta}{2} \mathbf{w}^\intercal \Sigma \mathbf{w}$ |
| multivariate Student | $\boldsymbol{\mu}, \boldsymbol{\Sigma}, \nu$ | does not exist |
| multivariate Laplace | $\boldsymbol{\mu}, \boldsymbol{\Sigma}$ | $-\frac{1}{\beta} \log(1 - \frac{\beta^2}{2} \mathbf{w}^\intercal \Sigma \mathbf{w})$ |
| multivariate logistic | $\boldsymbol{\mu}, \boldsymbol{\Sigma}$ | $\frac{1}{\beta} \log \mathrm{B}(1 - \beta \sqrt{\mathbf{w}^\intercal \Sigma \mathbf{w}}, 1 + \beta \sqrt{\mathbf{w}^\intercal \Sigma \mathbf{w}})$ |

Table 2: Table of common elliptical distributions with corresponding variance penalties. Here B denotes the Beta function.

**Skew-Elliptical Distributions.** The elliptical distributions represent a well-known class of symmetric probability distributions, containing both heavy-tailed and light-tailed members as special cases. However, they cannot capture skew in the return distribution that often arises in strictly discounted MDPs. The class of *generalized skew-elliptical distributions* (GSE) can model skew by extending the characteristic function (22) to

$$\mathbb{E}[e^{i\mathbf{t}^\intercal X}] = e^{i\mathbf{t}^\intercal \boldsymbol{\mu}} \xi(\mathbf{t}^\intercal \Sigma \mathbf{t}) k_d(\mathbf{t}), \quad \forall \mathbf{t} \in \mathbb{R}^d, \tag{24}$$

where $k_d : \mathbb{R}^d \to \mathbb{R}$ is some positive-valued function. In this case, we write $X \sim \mathcal{SE}_d(\boldsymbol{\mu}, \Sigma, \xi)$.

As for the elliptical distributions (Lemma 2), it is possible to show that GSE distributions are also closed under affine transforms [59].

**Lemma 3.** *Let* $X \sim \mathcal{SE}_d(\boldsymbol{\mu}, \Sigma, \xi)$ *and* $\mathbf{w} \in \mathbb{R}^d$. *Then,* $X^\intercal \mathbf{w}$ *is univariate GSE with characteristic function* $\mathbb{E}[e^{itX^\intercal \mathbf{w}}] = e^{itX^\intercal \boldsymbol{\mu}} \xi\left(t^2 \mathbf{w}^\intercal \Sigma \mathbf{w}\right) k(t; \mathbf{w}, \Sigma)$ *for some real-valued function* $k$.

By using Lemma 3 and the substitution $t = -i\beta$,

$$U_\beta[\Psi_h^\pi(s,a)^\intercal \mathbf{w}] = \boldsymbol{\psi}_h^\pi(s,a)^\intercal \mathbf{w} + \frac{1}{\beta} \log \xi\left(-\beta^2 \mathbf{w}^\intercal \Sigma_h^\pi(s,a) \mathbf{w}\right) + \frac{1}{\beta} \log k(-i\beta; \mathbf{w}, \Sigma_h^\pi(s,a)).$$

This new approximation generalizes (23) through the introduction of the term $\log k$, which intuitively captures the skew of the return distribution.

**Mixtures Densities.** One significant limitation of elliptical (and skew-elliptical) distributions to model returns is that they are unimodal, and can fail to capture multimodal risks in the environment. Consider the following *mixture of elliptical distributions* on $\mathbb{R}^d$:

$$I \sim \mathrm{Categorical}_K(\boldsymbol{\pi}),$$
$$X \mid I = k \sim \mathcal{E}_d(\boldsymbol{\mu}_k, \Sigma_k, \xi_k),$$

where $\boldsymbol{\pi} \in \mathbb{R}^K$ satisfies $\pi_k \geq 0$ and $\sum_k \pi_k = 1$ and $k = 1, \dots K$ define the possible modes of the distribution. In other words, each component of the mixture is a member of an elliptical distribution. This model extends the standard Gaussian mixture model, and can approximate any continuous distribution to arbitrary accuracy provided $K$ is chosen sufficiently large [58, 61]. In the context of risk-aware transfer (Theorem 1, Theorem 2) this means that the approximation error terms in $\varepsilon$ associated with approximating $U_\beta$ could in principle be driven to zero.

Applying Lemma 2 to each component, $X^\intercal \mathbf{w} \mid I = k \sim \mathcal{E}_1(\boldsymbol{\mu}_k^\intercal \mathbf{w}, \mathbf{w}^\intercal \Sigma_k \mathbf{w}, \xi_k)$, and so $X^\intercal \mathbf{w}$ is a mixture of univariate elliptical distributions. Now, (6, 12) can be computed using the law of total expectation,

$$U_\beta[\Psi_h^\pi(s,a)^\intercal \mathbf{w}] = \frac{1}{\beta} \log \sum_{k=1}^K \pi_k^\pi(s,a) e^{\beta \psi_k^\pi(s,a)^\intercal \mathbf{w}} \xi_k(-\beta^2 \mathbf{w}^\intercal \Sigma_k^\pi(s,a) \mathbf{w}).$$

This expression does not simplify further unless $K = 1$ and is thus *not* a mean-variance approximation, although it *is* a generalization of (23). It can be computed numerically by using the log-sum-exp trick, and the parameters of its associated mixture density could be learned in the Bellman framework using expectation propagation [56].

# B  Proofs of Theoretical Results

In this section, we verify all the theoretical claims stated in the main paper for the episodic MDPs. For ease of exposition and due to space limitations, we also state and prove similar results for discounted MDPs in this section[5].

## B.1  Proof of Lemma 1

**Lemma 1.** *Let $\beta \in \mathbb{R}$ and $X, Y$ be arbitrary random variables on $\Omega$. Then:*

> **A1** *(monotonicity) if $\mathbb{P}(X \geq Y) = 1$ then $U_\beta[X] \geq U_\beta[Y]$*
> **A2** *(cash invariance) $U_\beta[X + c] = U_\beta[X] + c$ for every $c \in \mathbb{R}$*
> **A3** *(convexity) if $\beta < 0$ ($\beta > 0$) then $U_\beta$ is a concave (convex) function*
> **A4** *(non-expansion) for $f, g : \Omega \rightarrow \Omega$, it follows that*
>
> $$|U_\beta[f(X)] - U_\beta[g(X)]| \leq \sup_{P \in \mathscr{P}_X(\Omega)} \mathbb{E}_P |f(X) - g(X)|,$$
>
> *where $\mathscr{P}_X(\Omega)$ is the set of all probability distributions on $\Omega$ that are absolutely continuous w.r.t. the true distribution of $X$.*

*Proof.* The first three properties are derived in Föllmer and Schied [13]. As for the fourth property, we use **A3** and convex duality [13], [54] to write

$$U_\beta[R] = \sup_{P \in \mathscr{P}_R(\Omega)} \left\{ \mathbb{E}_P[R] - \frac{1}{\beta} D(P\|P^*) \right\}, \tag{25}$$

where $\mathscr{P}_R(\Omega)$ is the set of all probability distributions on $\Omega$ absolutely continuous w.r.t. the true distribution $P^*$ of $R$, $D$ is the KL-divergence between $P$ and $P^*$. Now, for $f, g : \Omega \rightarrow \Omega$, $f$ and $g$ are bounded and hence $P$-integrable for any $P \in \mathscr{P}_R(\Omega)$, and using (25):

$$|U_\beta[f(X)] - U_\beta[g(X)]|$$
$$= \left| \sup_{P \in \mathscr{P}_X(\Omega)} \left\{ \mathbb{E}_P[f(X)] - \frac{1}{\beta} D(P\|P^*) \right\} - \sup_{P \in \mathscr{P}_X(\Omega)} \left\{ \mathbb{E}_P[g(X)] - \frac{1}{\beta} D(P\|P^*) \right\} \right|$$
$$\leq \sup_{P \in \mathscr{P}_X(\Omega)} |\mathbb{E}_P[f(X)] - \mathbb{E}_P[g(X)]|$$
$$\leq \sup_{P \in \mathscr{P}_X(\Omega)} \mathbb{E}_P |f(X) - g(X)|.$$

This completes the proof. $\qquad\qquad\square$

## B.2  Proofs of Theorem 1 and 2 for Episodic MDPs

**Theorem 1.** *Let $\pi_1, \ldots \pi_n$ be arbitrary deterministic Markov policies with approximate entropic utilities $\tilde{\mathcal{Q}}_{h,\beta}^{\pi_1}, \ldots \tilde{\mathcal{Q}}_{h,\beta}^{\pi_n}$ evaluated in an arbitrary task $M$, such that the errors satisfy $|\tilde{\mathcal{Q}}_{h,\beta}^{\pi_i}(s, a) - \mathcal{Q}_{h,\beta}^{\pi_i}(s, a)| \leq \varepsilon$ for all $s \in \mathcal{S}$, $a \in \mathcal{A}$, $i = 1 \ldots n$ and $h \in \mathcal{T}$. Define*

$$\pi_h(s) \in \underset{a \in \mathcal{A}}{\arg\max} \, \max_{i=1\ldots n} \tilde{\mathcal{Q}}_{h,\beta}^{\pi_i}(s, a), \quad \forall s \in \mathcal{S}.$$

*Then,*

$$\mathcal{Q}_{h,\beta}^{\pi}(s, a) \geq \max_i \mathcal{Q}_{h,\beta}^{\pi_i}(s, a) - 2(T - h + 1)\varepsilon, \quad h \leq T.$$

*Proof.* We have for all $h$ that

$$|\max_i \mathcal{Q}_h^{\pi_i}(s, a) - \max_i \tilde{\mathcal{Q}}_h^{\pi_i}(s, a)| \leq \max_i |\mathcal{Q}_h^{\pi_i}(s, a) - \tilde{\mathcal{Q}}_h^{\pi_i}(s, a)| \leq \varepsilon$$

---

[5]While the analysis of GPI in the risk-neutral setting [2] follows Strehl and Littman [60], our analysis of risk-aware GPI is also inspired by Huang and Haskell [53].

We proceed by induction on $h$. Clearly, the desired result holds for $h = T + 1$ since $\mathcal{Q}^{\pi_i}_{T+1,\beta}(s,a) = \tilde{\mathcal{Q}}^{\pi_i}_{T+1,\beta}(s,a) = 0$ uniformly. Next, suppose that $\mathcal{Q}^{\pi}_{h+1,\beta}(s,a) \geq \max_i \mathcal{Q}^{\pi_i}_{h+1,\beta}(s,a) - 2\varepsilon(T-h)$ holds uniformly at time $h+1$. Using **A1** and **A2** of Lemma 1:

$$
\begin{aligned}
\mathcal{Q}^{\pi}_{h,\beta}(s,a) &= U_\beta[r(s,a,s') + \mathcal{Q}^{\pi}_{h+1,\beta}(s',\pi_{h+1}(s'))] \\
&\geq U_\beta[r(s,a,s') + \max_i \mathcal{Q}^{\pi_i}_{h+1,\beta}(s',\pi_{h+1}(s')) - 2\varepsilon(T-h)] \\
&\geq U_\beta[r(s,a,s') + \max_i \tilde{\mathcal{Q}}^{\pi_i}_{h+1,\beta}(s',\pi_{h+1}(s')) - 2\varepsilon(T-h) - \varepsilon] \\
&= U_\beta[r(s,a,s') + \max_{a'} \max_i \tilde{\mathcal{Q}}^{\pi_i}_{h+1,\beta}(s',a') - 2\varepsilon(T-h) - \varepsilon] \\
&\geq U_\beta[r(s,a,s') + \max_{a'} \max_i \mathcal{Q}^{\pi_i}_{h+1,\beta}(s',a') - 2\varepsilon(T-h+1)] \\
&\geq U_\beta[r(s,a,s') + \max_i \max_{a'} \mathcal{Q}^{\pi_i}_{h+1,\beta}(s',a')] - 2\varepsilon(T-h+1) \\
&\geq U_\beta[r(s,a,s') + \max_i \mathcal{Q}^{\pi_i}_{h+1,\beta}(s',\pi_{i,h+1}(s'))] - 2\varepsilon(T-h+1) \\
&\geq U_\beta[r(s,a,s') + \mathcal{Q}^{\pi_i}_{h+1,\beta}(s',\pi_{i,h+1}(s'))] - 2\varepsilon(T-h+1) \\
&= \mathcal{Q}^{\pi_i}_{h,\beta}(s,a) - 2\varepsilon(T-h+1).
\end{aligned}
$$

Since $i$ is arbitrary, the proof is complete. $\qquad\square$

**Lemma 4.** *Let $\mathcal{Q}^{ij}_h$ be the utility of policy $\pi^*_i$ evaluated in task $j$ at time $h$. Then for all $i,j$,*

$$
\sup_{s,a} \left| \mathcal{Q}^{ii}_h(s,a) - \mathcal{Q}^{jj}_h(s,a) \right| \leq (T-h+1)\delta_{ij}.
$$

*Proof.* Let $\Delta_{ij}(h) = \sup_{s,a} |\mathcal{Q}^{ii}_h(s,a) - \mathcal{Q}^{jj}_h(s,a)|$. Define $\mathscr{P}_{s,a}$ to be the set of probability distributions that are absolutely continuous with respect to $P(\cdot|s,a)$. Then, using **A4** from Lemma 1:

$$
\begin{aligned}
&\Delta_{ij}(h) \\
&= \sup_{s,a} \left| \mathcal{Q}^{ii}_h(s,a) - \mathcal{Q}^{jj}_h(s,a) \right| \\
&= \sup_{s,a} \left| U_\beta[r_i(s,a,s') + \max_{a'} \mathcal{Q}^{ii}_{h+1}(s',a')] - U_\beta[r_j(s,a,s') + \max_{a'} \mathcal{Q}^{jj}_{h+1}(s',a')] \right| \\
&\leq \sup_{s,a} \sup_{P' \in \mathscr{P}_{s,a}} \mathbb{E}_{s' \sim P'(\cdot|s,a)} \left| r_i(s,a,s') - r_j(s,a,s') + \max_{a'} \mathcal{Q}^{ii}_{h+1}(s',a') - \max_{a'} \mathcal{Q}^{jj}_{h+1}(s',a') \right| \\
&\leq \sup_{s,a} \sup_{s'} \left| r_i(s,a,s') - r_j(s,a,s') + \max_{a'} \mathcal{Q}^{ii}_{h+1}(s',a') - \max_{a'} \mathcal{Q}^{jj}_{h+1}(s',a') \right| \\
&\leq \sup_{s,a,s'} |r_i(s,a,s') - r_j(s,a,s')| + \sup_{s,a,s'} \left| \max_{a'} \mathcal{Q}^{ii}_{h+1}(s',a') - \max_{a'} \mathcal{Q}^{jj}_{h+1}(s',a') \right| \\
&= \delta_{ij} + \sup_{s,a} \left| \mathcal{Q}^{ii}_{h+1}(s,a) - \mathcal{Q}^{jj}_{h+1}(s,a) \right| \\
&= \delta_{ij} + \Delta_{ij}(h+1).
\end{aligned}
$$

Starting with $\Delta_{ij}(T+1) = 0$ and proceeding by backward induction, we have $\Delta_{ij}(h) \leq \delta_{ij}(T-h+1)$ for all $h$. $\qquad\square$

**Lemma 5.**
$$
\sup_{s,a} \left| \mathcal{Q}^{jj}_h(s,a) - \mathcal{Q}^{ji}_h(s,a) \right| \leq (T-h+1)\delta_{ij}.
$$

*Proof.* Define $\Gamma_{ij}(h) = \sup_{s,a} |\mathcal{Q}^{jj}_h(s,a) - \mathcal{Q}^{ji}_h(s,a)|$. Then using **A4** from Lemma 1:

$$
\begin{aligned}
&\Gamma_{ij}(h) \\
&= \sup_{s,a} \left| \mathcal{Q}^{jj}_h(s,a) - \mathcal{Q}^{ji}_h(s,a) \right|
\end{aligned}
$$

$$= \sup_{s,a} \left| U_\beta[r_j(s,a,s') + \mathcal{Q}_{h+1}^{jj}(s', \pi_{j,h+1}^*(s'))] - U_\beta[r_i(s,a,s') + \mathcal{Q}_{h+1}^{ji}(s', \pi_{j,h+1}^*(s'))] \right|$$

$$\leq \sup_{s,a} \sup_{P' \in \mathscr{P}_{s,a}} \mathbb{E}_{s' \sim P'(\cdot|s,a)} \left| r_i(s,a,s') - r_j(s,a,s') + \mathcal{Q}_{h+1}^{jj}(s', \pi_{j,h+1}^*(s')) - \mathcal{Q}_{h+1}^{ji}(s', \pi_{j,h+1}^*(s')) \right|$$

$$\leq \sup_{s,a} \sup_{s'} \left| r_i(s,a,s') - r_j(s,a,s') + \mathcal{Q}_{h+1}^{jj}(s', \pi_{j,h+1}^*(s')) - \mathcal{Q}_{h+1}^{ji}(s', \pi_{j,h+1}^*(s')) \right|$$

$$\leq \sup_{s,a,s'} \left| r_i(s,a,s') - r_j(s,a,s') \right| + \sup_{s,a,s'} \left| \mathcal{Q}_{h+1}^{jj}(s', \pi_{j,h+1}^*(s')) - \mathcal{Q}_{h+1}^{ji}(s', \pi_{j,h+1}^*(s')) \right|$$

$$\leq \delta_{ij} + \sup_{s,a} \left| \mathcal{Q}_{h+1}^{jj}(s,a) - \mathcal{Q}_{h+1}^{ji}(s,a) \right|$$

$$= \delta_{ij} + \Gamma_{ij}(h+1).$$

Thus, $\Gamma_{ij}(h) \leq \delta_{ij}(T - h + 1)$ as claimed. $\qquad\square$

**Theorem 2.** *Let $\mathcal{Q}_{h,\beta}^{\pi_i^*}$ be the utilities of optimal Markov policies $\pi_i^*$ evaluated in task $M$. Furthermore, let $\tilde{\mathcal{Q}}_{h,\beta}^{\pi_i^*}$ be such that $|\tilde{\mathcal{Q}}_{h,\beta}^{\pi_i^*}(s,a) - \mathcal{Q}_{h,\beta}^{\pi_i^*}(s,a)| < \varepsilon$ for all $s \in \mathcal{S}, a \in \mathcal{A}$, $h \in \mathcal{T}$ and $i = 1 \ldots n$. Similarly, let $\pi$ be the corresponding policy in (4). Finally, let $\delta_r = \min_{i=1\ldots n} \sup_{s,a,s'} |r(s,a,s') - r_i(s,a,s')|$. Then,*

$$\left| \mathcal{Q}_{h,\beta}^\pi(s,a) - \mathcal{Q}_{h,\beta}^*(s,a) \right| \leq 2(T - h + 1)(\delta_r + \varepsilon), \quad h \leq T.$$

*Proof.* Using Theorem 1 and the triangle inequality:

$$\left| \mathcal{Q}_{h,\beta}^\pi(s,a) - \mathcal{Q}_{h,\beta}^*(s,a) \right| \leq \left| \mathcal{Q}_{h,\beta}^{\pi_j^*}(s,a) - \mathcal{Q}_{h,\beta}^*(s,a) \right| + 2(T - h + 1)\varepsilon.$$

The goal now is to bound the first term. By the triangle inequality and Lemma 4 and Lemma 5, $|\mathcal{Q}_h^{ii}(s,a) - \mathcal{Q}_h^{ji}(s,a)| \leq |\mathcal{Q}_h^{ii}(s,a) - \mathcal{Q}_h^{jj}(s,a)| + |\mathcal{Q}_h^{jj}(s,a) - \mathcal{Q}_h^{ji}(s,a)| = 2(T - h + 1)\delta_{ij}$. Finally, designating $j$ as source task $j$ and $i$ as target task, and substituting this bound into the first inequality above yields the desired result. $\qquad\square$

### B.3 Proofs of Theorem 1 and 2 for Discounted MDPs

**Theorem 5.** *Let $\pi_1, \ldots \pi_n$ be arbitrary deterministic Markov policies with approximate entropic utilities $\tilde{\mathcal{J}}_\beta^{\pi_1}, \ldots \tilde{\mathcal{J}}_\beta^{\pi_n}$ evaluated in an arbitrary task $M$, such that the errors satisfy $|\tilde{\mathcal{J}}_\beta^{\pi_i}(s,a,z) - \mathcal{J}_\beta^{\pi_i}(s,a,z)| \leq \varepsilon z$ for all $s, a, z$ and $i$. Define*

$$\pi(s,z) \in \arg\max_{a \in \mathcal{A}} \max_{i=1\ldots n} \tilde{\mathcal{J}}_\beta^{\pi_i}(s,a,z), \quad \forall s \in \mathcal{S}, z \in \mathcal{Z}. \tag{26}$$

*Then,*

$$\mathcal{J}_\beta^\pi(s,a,z) \geq \max_i \mathcal{J}_\beta^{\pi_i}(s,a,z) - \frac{2\varepsilon}{1-\gamma} z.$$

*Proof.* Define $\mathcal{J}_\beta^{max}(s,a,z) = \max_i \mathcal{J}_\beta^{\pi_i}(s,a,z)$ and $\tilde{\mathcal{J}}_\beta^{max}(s,a,z) = \max_i \tilde{\mathcal{J}}_\beta^{\pi_i}(s,a,z)$. We have:

$$|\mathcal{J}_\beta^{max}(s,a,z) - \tilde{\mathcal{J}}_\beta^{max}(s,a,z)| \leq \max_i |\mathcal{J}_\beta^{\pi_i}(s,a,z) - \tilde{\mathcal{J}}_\beta^{\pi_i}(s,a,z)| \leq \varepsilon z.$$

Let $T_\beta^\pi$ be the operator corresponding to (11). Then using **A1** and **A2** of Lemma 1 leads to:

$$T_\beta^\pi \tilde{\mathcal{J}}_\beta^{max}(s,a,z) = U_\beta \left[ zr(s,a,s') + \tilde{\mathcal{J}}_\beta^{max}(s', \pi(s', \gamma z), \gamma z) \right]$$

$$= U_\beta \left[ zr(s,a,s') + \max_{a'} \tilde{\mathcal{J}}_\beta^{max}(s', a', \gamma z) \right]$$

$$\geq U_\beta \left[ zr(s,a,s') + \max_{a'} \mathcal{J}_\beta^{max}(s', a', \gamma z) \right] - \gamma\varepsilon z$$

$$\geq U_\beta \left[ zr(s,a,s') + \mathcal{J}_\beta^{max}(s', \pi_i(s', \gamma z), \gamma z) \right] - \gamma\varepsilon z$$

$$\geq U_\beta \left[ zr(s,a,s') + \mathcal{J}_\beta^{\pi_i}(s', \pi_i(s', \gamma z), \gamma z) \right] - \gamma \varepsilon z$$
$$= T_\beta^{\pi_i} \mathcal{J}_\beta^{\pi_i}(s,a,z) - \gamma \varepsilon z$$
$$= \mathcal{J}_\beta^{\pi_i}(s,a,z) - \gamma \varepsilon z$$
$$\geq \max_i \mathcal{J}_\beta^{\pi_i}(s,a,z) - \gamma \varepsilon z$$
$$\geq \tilde{\mathcal{J}}_\beta^{max}(s,a,z) - \varepsilon z - \gamma \varepsilon z$$

Finally, using **A1** of Lemma 1 and the fact that $T_\beta^\pi$ has a unique fixed point [5]:

$$\mathcal{J}_\beta^\pi(s,a,z) = \lim_{k \to \infty} (T_\beta^\pi)^k \tilde{\mathcal{J}}_\beta^{max}(s,a,z) \geq \tilde{\mathcal{J}}_\beta^{max}(s,a,z) - (1+\gamma)\frac{\varepsilon}{1-\gamma}z$$

$$\geq \mathcal{J}_\beta^{max}(s,a,z) - \frac{2\varepsilon}{1-\gamma}z,$$

and is the desired result. □

**Lemma 6.** *Define $\mathcal{J}_j^i(s,a,z)$ be the utility of optimal policy $\pi_i^*$ on the augmented MDP $i$ when evaluated in the augmented MDP $j$. Furthermore, let $\delta_{ij} = \sup_{s,a,s'} |r_i(s,a,s') - r_j(s,a,s')|$. Then,*

$$\sup_{s,a} \left| \mathcal{J}_i^i(s,a,z) - \mathcal{J}_j^j(s,a,z) \right| \leq \frac{\delta_{ij}}{1-\gamma}z.$$

*Proof.* Define $\Delta_{ij}(z) = \sup_{s,a} \left| \mathcal{J}_i^i(s,a,z) - \mathcal{J}_j^j(s,a,z) \right|$. Let $\mathscr{P}_{s,a}$ be the set of probability distributions for the one-step transitions of the augmented MDP, e.g. $P((s',z')|(s,z),a)$ that are absolutely continuous w.r.t. the true distribution. Since $P((s',z')|(s,z),a) = P(s'|s,a)\delta_{z\gamma}(z')$, and $\delta_{z\gamma}(z')$ is absolutely continuous only w.r.t. itself, the set $\mathscr{P}_{s,a}$ consists of all products $P(s'|s,a)\delta_{z\gamma}(z')$, where $P(s'|s,a)$ is absolutely continuous w.r.t. the true dynamics of the original MDP.

Now, using **A4** from Lemma 1:

$$\Delta_{ij}(z)$$
$$= \sup_{s,a} \left| \mathcal{J}_i^i(s,a,z) - \mathcal{J}_j^j(s,a,z) \right|$$
$$= \sup_{s,a} \left| U_\beta[zr_i(s,a,s') + \max_{a'} \mathcal{J}_i^i(s',a',\gamma z)] - U_\beta[zr_j(s,a,s') + \max_{a'} \mathcal{J}_j^j(s',a',\gamma z)] \right|$$
$$\leq \sup_{s,a} \sup_{P \in \mathscr{P}_{s,a}} \mathbb{E}_{s' \sim P(\cdot|s,a)} \left| zr_i(s,a,s') - zr_j(s,a,s') + \max_{a'} \mathcal{J}_i^i(s',a',\gamma z) - \max_{a'} \mathcal{J}_j^j(s',a',\gamma z) \right|$$
$$\leq z \sup_{s,a,s'} |r_i(s,a,s') - r_j(s,a,s')| + \sup_{s,a,s'} \left| \max_{a'} \mathcal{J}_i^i(s',a',\gamma z) - \max_{a'} \mathcal{J}_j^j(s',a',\gamma z) \right|$$
$$= z\delta_{ij} + \sup_{s,a} \left| \mathcal{J}_i^i(s,a,\gamma z) - \mathcal{J}_j^j(s,a,\gamma z) \right|$$
$$= z\delta_{ij} + \Delta_{ij}(\gamma z).$$

Repeating the above bounding procedure leads to:

$$\Delta_{ij}(z) \leq z\delta_{ij} + \Delta_{ij}(\gamma z)$$
$$\leq z\delta_{ij} + \gamma z\delta_{ij} + \Delta_{ij}(\gamma^2 z)$$
$$\vdots$$
$$\leq z\delta_{ij} + \gamma z\delta_{ij} + \gamma^2 z\delta_{ij} + \cdots = \frac{\delta_{ij}}{1-\gamma}z,$$

and completes the proof. □

**Lemma 7.**

$$\sup_{s,a} \left| \mathcal{J}_j^j(s,a,z) - \mathcal{J}_i^j(s,a,z) \right| \leq \frac{\delta_{ij}}{1-\gamma}z.$$

*Proof.* Define $\Gamma_{ij}(z) = \sup_{s,a} |\mathcal{J}_j^j(s,a,z) - \mathcal{J}_i^j(s,a,z)|$. Then, using **A4** from Lemma 1 and the technique from Lemma 6:

$$\Gamma_{ij}(z)$$

$$= \sup_{s,a} \left| \mathcal{J}_j^j(s,a,z) - \mathcal{J}_i^j(s,a,z) \right|$$

$$= \sup_{s,a} \left| U_\beta[zr_j(s,a,s') + \mathcal{J}_j^j(s',\pi_j^*(s',\gamma z),\gamma z)] - U_\beta[zr_i(s,a,s') + \mathcal{J}_i^j(s',\pi_j^*(s',\gamma z),\gamma z)] \right|$$

$$\leq \sup_{s,a} \sup_{P \in \mathscr{P}_{s,a}} \mathbb{E}_{s' \sim P(\cdot|s,a)} \left| zr_i(s,a,s') - zr_j(s,a,s') + \mathcal{J}_j^j(s',\pi_j^*(s',\gamma z),\gamma z) - \mathcal{J}_i^j(s',\pi_j^*(s',\gamma z),\gamma z) \right|$$

$$\leq z \sup_{s,a,s'} |r_i(s,a,s') - r_j(s,a,s')| + \sup_{s,a,s'} \left| \mathcal{J}_j^j(s',\pi_j^*(s',\gamma z),\gamma z) - \mathcal{J}_i^j(s',\pi_j^*(s',\gamma z),\gamma z) \right|$$

$$\leq z\delta_{ij} + \sup_{s,a} \left| \mathcal{J}_j^j(s,a,\gamma z) - \mathcal{J}_i^j(s,a,\gamma z) \right|$$

$$= z\delta_{ij} + \Gamma_{ij}(\gamma z)$$

$$\leq \frac{\delta_{ij}}{1-\gamma} z.$$

The proof is complete. $\qquad\square$

**Theorem 6.** *Let $\mathcal{J}_\beta^{\pi_i^*}$ be the utilities of optimal Markov policies $\pi_i^*$ evaluated in some task $M$. Furthermore, let $\tilde{\mathcal{J}}_\beta^{\pi_i^*}$ be such such that $|\tilde{\mathcal{J}}_\beta^{\pi_i^*}(s,a,z) - \mathcal{J}_\beta^{\pi_i^*}(s,a,z)| < \varepsilon z$ for all $s \in \mathcal{S}$, $a \in \mathcal{A}$, $z \in \mathcal{Z}$ and $i = 1 \ldots n$, and $\pi$ be the corresponding policy in (26). Finally, let $\delta_r = \min_{i=1 \ldots n} \sup_{s,a,s'} |r(s,a,s') - r_i(s,a,s')|$. Then,*

$$\left| \mathcal{J}_\beta^\pi(s,a,z) - \mathcal{J}_\beta^*(s,a,z) \right| \leq \frac{2(\delta_r + \varepsilon)}{1-\gamma} z.$$

*Proof.* Using Theorem 5:

$$\mathcal{J}_\beta^\pi(s,a,z) - \mathcal{J}_\beta^*(s,a,z) = \mathcal{J}_\beta^\pi(s,a,z) - \mathcal{J}_\beta^{\pi_j^*}(s,a,z) + \mathcal{J}_\beta^{\pi_j^*}(s,a,z) - \mathcal{J}_\beta^*(s,a,z)$$

$$\geq \frac{2\varepsilon}{1-\gamma} z + \mathcal{J}_\beta^{\pi_j^*}(s,a,z) - \mathcal{J}_\beta^*(s,a,z).$$

The goal now is to bound the difference between the last two terms. Let $\mathcal{J}_j^i(s,a)$ be the entropic utility of the optimal policy $\pi_i^*$ evaluated in the augmented MDP for task $j$. Then, by the triangle inequality, $|\mathcal{J}_i^i(s,a,z) - \mathcal{J}_j^j(s,a,z)| \leq |\mathcal{J}_i^i(s,a,z) - \mathcal{J}_j^j(s,a,z)| + |\mathcal{J}_j^j(s,a,z) - \mathcal{J}_i^j(s,a,z)|$. Applying Lemma 6 and Lemma 7, we have $|\mathcal{J}_i^i(s,a,z) - \mathcal{J}_j^j(s,a,z)| \leq \frac{2\delta_{ij}}{1-\gamma} z$, and the result follows. $\qquad\square$

### B.4 Proof of Theorem 3

The proofs depend on the following result adapted from Sherstan et al. [37].

**Lemma 8.** *Let $X$ be a random vector in $\mathbb{R}^d$ that depends only on $s_h, a_h, r_h$ and $s_{h+1}$. Then,*

$$\mathbb{E}\left[ X(\Psi_{h+1}^\pi(s',\pi_{h+1}(s')) - \psi_{h+1}^\pi(s',\pi_{h+1}(s')))^\mathsf{T} \mid s_h = s, a_h = a \right] = 0.$$

We first demonstrate that the Bellman equation (8) is correct for our problem.

**Lemma 9.**

$$\Sigma_h^\pi(s,a) = \mathbb{E}_{s' \sim P(\cdot|s,a)} \left[ \boldsymbol{\delta}_h \boldsymbol{\delta}_h^\mathsf{T} + \Sigma_{h+1}^\pi(s',\pi_{h+1}(s')) \mid s_h = s, a_h = a \right].$$

*Proof.* Let $\Psi_h^\pi(s,a) = \phi_h + \phi_{h+1} + \ldots$ and define $\boldsymbol{\xi}_h^\pi(s,a) = \Psi_h^\pi(s,a) - \psi_h^\pi(s,a)$. By definition of successor features, we have:

$$\boldsymbol{\xi}_h^\pi(s,a) = \Psi_h^\pi(s,a) - \psi_h^\pi(s,a)$$

$$\begin{aligned}
&= \boldsymbol{\phi}_h + \boldsymbol{\psi}_{h+1}^\pi(s', \pi_{h+1}(s')) - \boldsymbol{\psi}_h^\pi(s, a) + (\Psi_{h+1}^\pi(s', \pi_{h+1}(s')) - \boldsymbol{\psi}_{h+1}^\pi(s', \pi_{h+1}(s'))) \\
&= \boldsymbol{\delta}_h + \boldsymbol{\xi}_{h+1}^\pi(s', \pi_{h+1}(s'))
\end{aligned}$$

By definition, the covariance is:

$$\begin{aligned}
\Sigma_h^\pi(s, a) &= \mathbb{E}\left[(\Psi_h^\pi(s, a) - \boldsymbol{\psi}_h^\pi(s, a))(\Psi_h^\pi(s, a) - \boldsymbol{\psi}_h^\pi(s, a))^\mathsf{T} \mid s_h = s,\, a_h = a\right] \\
&= \mathbb{E}\left[\boldsymbol{\xi}_h^\pi(s, a)\boldsymbol{\xi}_h^\pi(s, a)^\mathsf{T} \mid s_h = s,\, a_h = a\right] \\
&= \mathbb{E}\left[(\boldsymbol{\delta}_h + \boldsymbol{\xi}_{h+1}^\pi(s', \pi_{h+1}(s')))(\boldsymbol{\delta}_h + \boldsymbol{\xi}_{h+1}^\pi(s', \pi_{h+1}(s')))^\mathsf{T} \mid s_h = s,\, a_h = a\right] \\
&= \mathbb{E}\left[\boldsymbol{\delta}_h\boldsymbol{\delta}_h{}^\mathsf{T} + \boldsymbol{\xi}_{h+1}^\pi(s', \pi_{h+1}(s'))\boldsymbol{\xi}_{h+1}^\pi(s', \pi_{h+1}(s'))^\mathsf{T} \mid s_h = s,\, a_h = a\right] \\
&\quad + \mathbb{E}\left[\boldsymbol{\delta}_h\boldsymbol{\xi}_{h+1}^\pi(s', \pi_{h+1}(s'))^\mathsf{T} \mid s_h = s,\, a_h = a\right] \\
&\quad + \mathbb{E}\left[\boldsymbol{\xi}_{h+1}^\pi(s', \pi_{h+1}(s'))\boldsymbol{\delta}_h{}^\mathsf{T} \mid s_h = s,\, a_h = a\right] \\
&= \mathbb{E}\left[\boldsymbol{\delta}_h\boldsymbol{\delta}_h{}^\mathsf{T} + \boldsymbol{\xi}_{h+1}^\pi(s', \pi_{h+1}(s'))\boldsymbol{\xi}_{h+1}^\pi(s', \pi_{h+1}(s'))^\mathsf{T} \mid s_h = s,\, a_h = a\right] \\
&= \mathbb{E}\left[\boldsymbol{\delta}_h\boldsymbol{\delta}_h{}^\mathsf{T} + \Sigma_{h+1}^\pi(s', \pi_{h+1}(s')) \mid s_h = s,\, a_h = a\right],
\end{aligned}$$

where the second-last line follows from Lemma 8. $\qquad\square$

**Theorem 3.** *Let* $\|\cdot\|$ *be a matrix-compatible norm, and suppose there exists* $\varepsilon : \mathcal{S} \times \mathcal{A} \times \mathcal{T} \to [0, \infty)$ *such that:*

1. $\|\tilde{\boldsymbol{\psi}}_h^\pi(s, a) - \boldsymbol{\psi}_h^\pi(s, a)\|^2 \le \varepsilon_h(s, a)$

2. $\left\|\mathbb{E}_{s' \sim P(\cdot|s,a)}[\tilde{\boldsymbol{\delta}}_h(\tilde{\boldsymbol{\psi}}_h^\pi(s', \pi_{h+1}(s')) - \boldsymbol{\psi}_h^\pi(s', \pi_{h+1}(s')))^\mathsf{T}]\right\| \le \varepsilon_h(s, a)$.

*Then,*

$$\left\|\Sigma_h^\pi(s, a) - \mathbb{E}_{s' \sim P(\cdot|s,a)}\left[\tilde{\boldsymbol{\delta}}_h\tilde{\boldsymbol{\delta}}_h^\mathsf{T} + \tilde{\Sigma}_{h+1}^\pi(s', \pi_{h+1}(s'))\right]\right\| \le 3\varepsilon_h(s, a).$$

*Proof.* We start by decomposing the true covariance matrix:

$$\begin{aligned}
\Sigma_h^\pi(s, a) &= \mathbb{E}\Big[(\Psi_h^\pi(s, a) - \tilde{\boldsymbol{\psi}}_h^\pi(s, a) + \boldsymbol{\psi}_h^\pi(s, a) - \boldsymbol{\psi}_h^\pi(s, a)) \\
&\qquad (\Psi_h^\pi(s, a) - \tilde{\boldsymbol{\psi}}_h^\pi(s, a) + \boldsymbol{\psi}_h^\pi(s, a) - \boldsymbol{\psi}_h^\pi(s, a))^\mathsf{T} \mid s_h = s,\, a_h = a\Big] \\
&= \mathbb{E}\left[(\Psi_h^\pi(s, a) - \tilde{\boldsymbol{\psi}}_h^\pi(s, a))(\Psi_h^\pi(s, a) - \tilde{\boldsymbol{\psi}}_h^\pi(s, a))^\mathsf{T} \mid s_h = s,\, a_h = a\right] \\
&\quad + (\tilde{\boldsymbol{\psi}}_h^\pi(s, a) - \boldsymbol{\psi}_h^\pi(s, a))(\tilde{\boldsymbol{\psi}}_h^\pi(s, a) - \boldsymbol{\psi}_h^\pi(s, a))^\mathsf{T} \\
&\quad + 2\mathbb{E}\left[\Psi_h^\pi(s, a) - \tilde{\boldsymbol{\psi}}_h^\pi(s, a) \mid s_h = s,\, a_h = a\right](\tilde{\boldsymbol{\psi}}_h^\pi(s, a) - \boldsymbol{\psi}_h^\pi(s, a))^\mathsf{T} \\
&= \mathbb{E}\left[(\Psi_h^\pi(s, a) - \tilde{\boldsymbol{\psi}}_h^\pi(s, a))(\Psi_h^\pi(s, a) - \tilde{\boldsymbol{\psi}}_h^\pi(s, a))^\mathsf{T} \mid s_h = s,\, a_h = a\right] \\
&\quad - (\tilde{\boldsymbol{\psi}}_h^\pi(s, a) - \boldsymbol{\psi}_h^\pi(s, a))(\tilde{\boldsymbol{\psi}}_h^\pi(s, a) - \boldsymbol{\psi}_h^\pi(s, a))^\mathsf{T}
\end{aligned}$$

where in the last step we use the identity $\mathbb{E}\left[\Psi_h^\pi(s, a) - \tilde{\boldsymbol{\psi}}_h^\pi(s, a) \mid s_h = s,\, a_h = a\right] = \mathbb{E}\left[\Psi_h^\pi(s, a) - \boldsymbol{\psi}_h^\pi(s, a) \mid s_h = s,\, a_h = a\right] + \boldsymbol{\psi}_h^\pi(s, a) - \tilde{\boldsymbol{\psi}}_h^\pi(s, a) = \boldsymbol{\psi}_h^\pi(s, a) - \tilde{\boldsymbol{\psi}}_h^\pi(s, a)$. Now, we define $\tilde{\boldsymbol{\xi}}_h^\pi(s, a) = \Psi_h^\pi(s, a) - \tilde{\boldsymbol{\psi}}_h^\pi(s, a)$, then follow the derivations in Lemma 9 to write the first term above as:

$$\begin{aligned}
&\mathbb{E}\left[\tilde{\boldsymbol{\xi}}_h^\pi(s, a)\tilde{\boldsymbol{\xi}}_h^\pi(s, a)^\mathsf{T} \mid s_h = s,\, a_h = a\right] \\
&= \mathbb{E}\left[\tilde{\boldsymbol{\delta}}_h\tilde{\boldsymbol{\delta}}_h^\mathsf{T} + \tilde{\boldsymbol{\xi}}_{h+1}^\pi(s', \pi_{h+1}(s'))\tilde{\boldsymbol{\xi}}_{h+1}^\pi(s', \pi_{h+1}(s'))^\mathsf{T} \mid s_h = s,\, a_h = a\right] \\
&\quad + \mathbb{E}\left[\tilde{\boldsymbol{\delta}}_h\tilde{\boldsymbol{\xi}}_{h+1}^\pi(s', \pi_{h+1}(s'))^\mathsf{T} \mid s_h = s,\, a_h = a\right] + \mathbb{E}\left[\tilde{\boldsymbol{\xi}}_{h+1}^\pi(s', \pi_{h+1}(s'))\tilde{\boldsymbol{\delta}}_h^\mathsf{T} \mid s_h = s,\, a_h = a\right] \\
&= \mathbb{E}\left[\tilde{\boldsymbol{\delta}}_h\tilde{\boldsymbol{\delta}}_h^\mathsf{T} + \tilde{\Sigma}_{h+1}^\pi(s', \pi_{h+1}(s')) \mid s_h = s,\, a_h = a\right] \\
&\quad + \mathbb{E}\left[\tilde{\boldsymbol{\delta}}_h\tilde{\boldsymbol{\xi}}_{h+1}^\pi(s', \pi_{h+1}(s'))^\mathsf{T} \mid s_h = s,\, a_h = a\right] + \mathbb{E}\left[\tilde{\boldsymbol{\xi}}_{h+1}^\pi(s', \pi_{h+1}(s'))\tilde{\boldsymbol{\delta}}_h^\mathsf{T} \mid s_h = s,\, a_h = a\right].
\end{aligned}$$

Finally, we norm bound the desired difference as follows:

$$\left\| \Sigma_h^\pi(s,a) - \mathbb{E}\left[ \tilde{\boldsymbol{\delta}}_h \tilde{\boldsymbol{\delta}}_h^\mathsf{T} + \tilde{\Sigma}_{h+1}^\pi(s', \pi_{h+1}(s')) \mid s_h = s,\, a_h = a \right] \right\|$$

$$\leq 2 \left\| \mathbb{E}\left[ \tilde{\boldsymbol{\delta}}_h \tilde{\boldsymbol{\xi}}_{h+1}^\pi(s', \pi_{h+1}(s'))^\mathsf{T} \mid s_h = s,\, a_h = a \right] \right\|$$
$$+ \left\| (\tilde{\boldsymbol{\psi}}_h^\pi(s,a) - \boldsymbol{\psi}_h^\pi(s,a))(\tilde{\boldsymbol{\psi}}_h^\pi(s,a) - \boldsymbol{\psi}_h^\pi(s,a))^\mathsf{T} \right\|$$

$$\leq 2 \left\| \mathbb{E}\left[ \tilde{\boldsymbol{\delta}}_h (\Psi_{h+1}^\pi(s', \pi_{h+1}(s')) - \boldsymbol{\psi}_{h+1}^\pi(s', \pi_{h+1}(s')))^\mathsf{T} \mid s_h = s,\, a_h = a \right] \right\|$$
$$+ 2 \left\| \mathbb{E}\left[ \tilde{\boldsymbol{\delta}}_h (\boldsymbol{\psi}_{h+1}^\pi(s', \pi_{h+1}(s')) - \tilde{\boldsymbol{\psi}}_{h+1}^\pi(s', \pi_{h+1}(s')))^\mathsf{T} \mid s_h = s,\, a_h = a \right] \right\|$$
$$+ \left\| (\tilde{\boldsymbol{\psi}}_h^\pi(s,a) - \boldsymbol{\psi}_h^\pi(s,a))(\tilde{\boldsymbol{\psi}}_h^\pi(s,a) - \boldsymbol{\psi}_h^\pi(s,a))^\mathsf{T} \right\|$$

$$\leq 2 \left\| \mathbb{E}\left[ \tilde{\boldsymbol{\delta}}_h (\boldsymbol{\psi}_{h+1}^\pi(s', \pi_{h+1}(s')) - \tilde{\boldsymbol{\psi}}_{h+1}^\pi(s', \pi_{h+1}(s')))^\mathsf{T} \mid s_h = s,\, a_h = a \right] \right\|$$
$$+ \left\| \tilde{\boldsymbol{\psi}}_h^\pi(s,a) - \boldsymbol{\psi}_h^\pi(s,a) \right\|^2$$
$$\leq 2\varepsilon_h(s,a) + \varepsilon_h(s,a) = 3\varepsilon_h(s,a).$$

This is the desired result. $\qquad\square$

## C    Experiment Details

In this section, we describe the setup of the domains discussed in the main paper in greater detail. We also provide detailed descriptions of baseline algorithms, as well as all hyper-parameters used and how they were selected.

### C.1    Motivating Example

**Domain Configuration.**    The motivating example is a 5-by-5 grid-world domain with discrete states and discrete actions described by the four possible directions of movement into an adjacent cell. The environment is made stochastic by introducing random action noise as follows. Desired actions are taken only with probability $0.8$, while the remaining time a (uniformly) random action is taken. Furthermore, transitions that would take the agent outside of the boundaries of the grid leave the agent in its current position. The cost structure is defined as follows. The goal state is terminal and provides a reward of $+20$. Each time step incurs a fixed penalty of $-1$, on top of any other rewards or costs incurred.

**Learning Source Policies and Utilities.**    To recover the properties of risk-aware and risk-neutral GPI claimed in the main text, we first learn the source policies $\pi_1$ and $\pi_2$ and their utilities $\mathcal{Q}_\beta^{\pi_1}$ and $\mathcal{Q}_\beta^{\pi_2}$ using a variant of the classic value iteration algorithm adapted to maximize the entropic utility (see Algorithm 3). We consider the non-discounted setting ($\gamma = 1$), and iterate until an absolute error less than $\varepsilon_{exit} = 10^{-12}$ is achieved between two consecutive iterations[6]. The two source policies are then recovered by acting greedily with respect to the learned utilities.

**Transfer Learning.**    In order to implement GPI, we evaluate these two resulting policies on the target task by adapting the iterative procedure in Algorithm 3 for *policy evaluation*. Essentially, line 10 of the algorithm is replaced by $r(s, a, s') + \gamma \mathcal{Q}(s', \pi(s'))$ for $\pi \in \{\pi_1, \pi_2\}$. We repeat this procedure twice to produce two sets of value functions: a set $\{\mathcal{Q}_0^{\pi_1}, \mathcal{Q}_0^{\pi_2}\}$ for $\beta = 0$[7] and a set $\{\mathcal{Q}_{-0.1}^{\pi_1}, \mathcal{Q}_{-0.1}^{\pi_2}\}$ for $\beta = -0.1$. The two GPI policies are then defined as $\pi_\beta(s) \in \arg\max_a \max_{i=1,2} \mathcal{Q}_\beta^{\pi_i}(s,a)$ for $\beta \in \{0, -0.1\}$. Finally, we generate the histogram of returns by simulating episodes of length $T = 35$, in which actions are selected from $\pi_\beta$, and computing the cumulative reward obtained on each episode.

---

[6]Please note that convergence of value iteration is guaranteed due to the existence of absorbing states and because the underlying MDP is ergodic.

[7]For $\beta = 0$, Algorithm 3 reduces to standard value iteration.

**Algorithm 3** Value Iteration for Entropic Utility Maximization

---

1: **Requires** $\varepsilon_{exit} > 0$, $\gamma \in [0, 1]$, $\beta \in \mathbb{R}$, $\langle \mathcal{S}, \mathcal{A}, r, P \rangle \in \mathcal{M}$
2: **for** $s \in \mathcal{S}$, $a \in \mathcal{A}$ **do** $\mathcal{Q}(s, a) \leftarrow 0$
3: **for** $n = 1, 2 \ldots \infty$ **do**
       \\\\ Update $\mathcal{Q}(s, a)$ for all state-action pairs
4:     **for** $s \in \mathcal{S}$, $a \in \mathcal{A}$ **do**
           \\\\ Perform one iteration of (11) with the greedy policy derived from $\mathcal{Q}$
5:         $\mathcal{Q}'(s, a) \leftarrow 0$
6:         **for** $s' \in \mathcal{S}$ **do**
7:             **if** $s'$ is terminal **then**
8:                 target $\leftarrow r(s, a, s')$
9:             **else**
10:                target $\leftarrow r(s, a, s') + \gamma \max_b \mathcal{Q}(s', b)$
11:            **end if**
12:            $\mathcal{Q}'(s, a) \leftarrow \mathcal{Q}'(s, a) + P(s'|s, a) \, e^{\beta \times \text{target}}$
13:        **end for**
14:        $\mathcal{Q}'(s, a) \leftarrow \frac{1}{\beta} \log \mathcal{Q}'(s, a)$
15:    **end for**
       \\\\ Check for convergence in utility values
16:    $\varepsilon \leftarrow \max_{s,a} |\mathcal{Q}'(s, a) - \mathcal{Q}(s, a)|$
17:    **if** $\varepsilon < \varepsilon_{exit}$ **then return** $\mathcal{Q}'$
       \\\\ If not converged, then continue with value iteration
18:    **for** $s \in \mathcal{S}$, $a \in \mathcal{A}$ **do** $\mathcal{Q}(s, a) \leftarrow \mathcal{Q}'(s, a)$
19: **end for**

---

## C.2   Four-Room

**State and Action Spaces.**   The four-room domain consists of a family of discrete-state discrete-action MDPs $\mathcal{M}$ defined as follows. The world is defined as a set of discrete cells arranged in a grid of dimensions 13-by-13, such that at each time instant, the agent occupies a specific cell with some $x$- and $y$-coordinates $(p_x, p_y) \in \{0, \ldots 12\}^2$. As the agent explores the space, it can collect objects belonging to one of 3 possible classes. While the initial positions of these objects remains fixed throughout the experiment, their existence is determined by whether or not they have already been collected by the agent in a given episode (the same object cannot be picked up multiple times in a given episode). In our configuration, there are 6 instances of objects belonging to each class, for a total of $n_o = 18$ collectible objects. Therefore, the state space $\mathcal{S} = \{0, 1\}^{n_o} \times \{0, \ldots 12\}^2$ consists of the concatenation of the agent's current position $(p_x, p_y)$ and a set of binary variables indicating whether or not each object has already been picked up by the agent. All objects are reset at the beginning of each episode. Actions are defined as $\mathcal{A} = \{\text{left}, \text{up}, \text{right}, \text{down}\}$ that move the agent to an adjacent cell in the corresponding direction. In the case that the destination cell lies outside the grid, then the agent remains in the current cell at the next time instant.

**Reward Function and Risk.**   The goal cell 'G' provides a fixed reward of $+1$ and immediately terminates the episode upon entry. The reward $r_c$ associated with each object class $c \in \{1, 2, 3\}$ is reset every time a new task begins, and is sampled from a uniform distribution on $[-1, +1]$. Occupying a trap cell that triggers at a particular time instant defines a failure, and is communicated to the agent by incurring a penalty of $-2$ and immediately terminating the episode. However, occupying a trap cell does not automatically guarantee a failure. Instead, a failure is only triggered with probability 0.05 independently at every time instant during which the agent occupies a trap cell. This additional reward stochasticity can be implemented without breaking the existing successor feature framework by introducing a fictitious terminal state $s_f$ to indicate failure, which is reached at random when in cells marked 'X'. This state augmentation induces a modified MDP with a deterministic reward of $-2$ on arrival to state $s_f$, whose associated transitions are stochastic in nature. Crucially, this state augmentation transformation applies uniformly to all task instances, and thus does not break our assumptions about $\mathcal{M}$. We use a discount factor of $\gamma = 0.95$.

**Features and Linear Reward Parameterization.** Exact state features $\phi(s, a, s')$ are provided directly to the agent. Specifically, we define $\phi_c(s, a, s')$ for every class of objects $c \in \{1, 2, 3\}$ to take the value 1 if the agent occupies a cell with an object of class $c$ in state $s'$ and 0 otherwise. Similarly, we define $\phi_g(s, a, s')$ to take the value 1 if $s'$ corresponds to the goal cell and 0 otherwise. Unlike Barreto et al. [2], the four-room domain also contains an additional failure state with non-zero reward, as described above, and this must also be incorporated into the SF representation. This can be done by defining $\phi_f(s, a, s')$ that takes the value 1 if $s'$ corresponds to the state $s_f$ and 0 otherwise[8]. The state features $\phi \in \mathbb{R}^5$ are then the concatenation of $\phi_c$, $\phi_g$ and $\phi_f$. These features are sparse, but can represent the reward functions of all possible task instances in $\mathcal{M}$ exactly. Finally, we define $\mathbf{w}_c = r_c$, $\mathbf{w}_g = 1$ and $\mathbf{w}_f = -2$, and it is now clear that $r(s, a, s') = \phi(s, a, s')^\mathsf{T}\mathbf{w}$ holds.

**Hyper-Parameters.** Each time a new task is created, a new $\tilde{\psi}^\pi$ and $\tilde{\Sigma}^\pi$ are created. The training loop of RaSFQL then proceeds according to Algorithm 1. We set $\alpha = 0.5$ and $\varepsilon = 0.12$, based on preliminary experiments for Q-learning. We also set $\bar{\alpha} = 0.1$ for learning $\tilde{\Sigma}^\pi$ and $\alpha_w = 0.5$ for learning $\mathbf{w}$ with gradient descent. Rollouts are limited to $T = 200$ steps for all algorithms.

**Baseline.** The baseline used for comparison is the *probabilistic policy reuse* framework of Fernández and Veloso [12] (PRQL), here adapted for learning risk-sensitive behaviors. In order to do this, we incorporate the *smart exploration* strategy of Gehring and Precup [16]. This strategy is fundamentally similar to our mean-variance approach, since it also incorporates second-moment or reward-variance information into action selection in a similar way. The controllability bonus $C^\pi(s, a)$ in each state-action pair is learned using a Q-learning approach by using the negative of the absolute Bellman residuals $-|\delta|$ as pseudo-rewards, and learned in parallel to the Q-values in practical implementations. The penalty for $C(s, a)$ is denoted as $\omega$, and is fundamentally similar to $\beta$ used by SFQL. The resulting algorithm, which we call RaPRQL, is described in Algorithm 4.

**Baseline Hyper-Parameters.** Similar to RaSFQL, every time a new task is created, a new $Q^\pi$ and $C^\pi$ are created for RaPRQL for learning new policies. We set $\alpha = 0.5$ for fair comparison with RaSFQL, and $\rho = 0.1$ based on the original implementation [16]. The performance is highly sensitive to the parameters $\eta$ and $\tau$ used by PRQL. To select these two hyper-parameters, we follow Barreto et al. [2] and run a grid search for $\eta \in \{0.1, 0.3, 0.5\}$ and $\tau \in \{1, 10, 100\}$, selecting the combination of $\eta$ and $\tau$ that resulted in the highest cumulative return over 128 task instances. This validation experiment is repeated for every value of $\omega$.

### C.3 Reacher

**State and Action Spaces.** The state space $\mathcal{S} \subset \mathbb{R}^4$ consists of the angles and angular velocities of the robotic arm's two joints. The two-dimensional action space $\mathcal{A} \subset [-1, +1]^2$ is discretized using 3 values per dimension, corresponding to maximum positive $(+1)$ and negative $(-1)$ and zero torque for each actuator, resulting in a total of 9 possible actions. At the beginning of each episode, the angle of the central joint is sampled from a uniform distribution on $[-\pi, +\pi]$, while the angle of the outer joint is sampled from a uniform distribution on $[-\pi/2, +\pi/2]$, and the angular velocities are initialized to zero. Furthermore, state transitions are made stochastic by adding zero-mean Gaussian noise to actions with standard deviation 0.03, and then clipping the actions to $[-1, +1]$.

**Reward Function and Risk.** The reward received at each time step is $1 - 4\delta$, where $\delta$ is the Euclidean distance between the target position and the tip of the robotic arm. We define 12 target locations, of which 4 are used for training and the remaining 8 for testing. Furthermore, circular regions of radius $\delta_f = 0.06$ are placed around 6 of the 12 target locations (2 training and 4 testing) in which failures occur spontaneously with probability $p_f = 0.035$. Once a failure occurs, a cost of $c_f = 3$ is incurred and the episode continues without termination. This implies that the expected

---

[8]It is not practical to redefine the task space with the augmented state $s_f$ in an actual implementation. Instead, we simulate this by providing the state features $\phi$ with a binary variable indicating failure. This does not change the SF implementation, since the occurrence of a failure event can be deduced using the done flag (indicating arrival in a terminal state) and the state $s'$.

---

**Algorithm 4** RaPRQL with Smart Exploration

---

1: **Requires** $m, T, N_e \in \mathbb{N}$, $\varepsilon, \eta \in [0, 1]$, $\alpha, \rho, \tau > 0$, $\omega \in \mathbb{R}$, $M_1, \dots M_m \in \mathcal{M}$
2: **for** $t = 1, 2 \dots m$ **do**
3:     Initialize $Q^t(s, a), C^t(s, a)$ to small random or zero values
4:     **for** $k = 1, 2 \dots t$ **do** $\text{score}_k \leftarrow 0$, $\text{used}_k \leftarrow 0$
5:     $c \leftarrow t$
6:     $R \leftarrow 0$
       \\ Commence training on task $M_t$
7:     **for** $n_e = 1, 2 \dots N_e$ **do**
8:         Initialize $M_t$ with initial state $s$
9:         **for** $h = 0, 1 \dots T$ **do**
            \\ Select actions according to Q-values plus controllability bonus
10:             **if** $c \neq t$ **then** use_prev_policy $\sim$ Bernoulli($\eta$) **else** use_prev_policy $\leftarrow$ false
11:             **if** use_prev_policy **then**
                \\ Action is selected from $\pi_c$, the source policy being used
12:                 $a \leftarrow \arg\max_b \{Q_h^c(s, b) + \omega C_h^c(s, b)\}$
13:             **else**
                \\ Action is selected from $\pi_t$, the policy being learned
14:                 random_a $\sim$ Bernoulli($\varepsilon$)
15:                 **if** random_a **then** $a \sim$ Uniform($\mathcal{A}$) **else** $a \leftarrow \arg\max_b \{Q_h^t(s, b) + \omega C_h^t(s, b)\}$
16:             **end if**
17:             Take action $a$ in $M_t$ and observe $r$ and $s'$
            \\ Update the Q-values for the current task
18:             $\delta_h \leftarrow r + \max_b Q_{h+1}^t(s', b) - Q_h^t(s, a)$
19:             $Q_h^t(s, a) \leftarrow Q_h^t(s, a) + \alpha \delta_h$
            \\ Update the controllability bonus for the current task
20:             $C_h^t(s, a) \leftarrow C_h^t(s, a) + \alpha\rho(-|\delta_h| - C_h^t(s, a))$
21:             $R \leftarrow R + r$
22:             $s \leftarrow s'$
23:         **end for**
        \\ Update average return obtained by following policy $\pi_c$
24:         $\text{score}_c \leftarrow \frac{\text{score}_c \times \text{used}_c + R}{\text{used}_c + 1}$
        \\ Sample a new source policy
25:         **for** $k = 1, 2 \dots t$ **do** $p_k \leftarrow \frac{e^{\tau \times \text{score}_k}}{\sum_j e^{\tau \times \text{score}_j}}$
26:         $c \sim$ Multinomial($p_1, p_2, \dots p_t$)
27:         $\text{used}_c \leftarrow \text{used}_c + 1$
28:         $R \leftarrow 0$
29:     **end for**
30: **end for**

---

reward, as a function of the distance $\delta$, is[9]

$$R(\delta) = \begin{cases} 1 - 4\delta & \text{if } \delta > \delta_f \\ 1 - 4\delta - c_f \times p_f & \text{if } \delta \leq \delta_f. \end{cases}$$

Therefore, a rational[10] risk-neutral agent would prefer to enter inside the failure region if it holds that $1 - c_f \times p_f \geq 1 - 4\delta_f$, or in other words if

$$c_f \times p_f \leq 4\delta_f.$$

Clearly, given our choice of values for $c_f, p_f$ and $\delta_f$, the above condition holds in our setting. Setting up the reward structure and risk in this way makes it possible to control the trade-off between risk and reward, and thus the anticipated behavior of the agents, in a principled way. We also apply discounting of future reward using $\gamma = 0.9$.

---

[9]The reasoning here has simplified some of the aspects of the environment, ignoring the effects of multiple risk regions that could alter the trajectories, limited-length episodes and discounting.

[10]Of course, a rational agent would want to keep the tip as close to the target location as possible, and so would want $\delta = 0$.

**Features and Linear Reward Parameterization.** The state features are vectors $\phi(s, a, s') \in \mathbb{R}^{13}$, in which the first 12 components consist of $1 - 4\delta_g$, where $\delta_g$ are the Euclidean distances to each of the goal locations $g$. The last component takes the value 1 if a failure event occurs and 0 otherwise. As done in the four-room experiment, state features are provided to the agent. However, target goal locations $\mathbf{w} \in \mathbb{R}^{13}$ are not learned in this instance, but provided directly to the agent as well. Specifically, we set $\mathbf{w}_g = 1$ for the goal with index $g$ and $\mathbf{w}_{13} = c_f = -3$, and set all other elements to zero. This recovers the correct reward function $r(s, a, s')$ for all task instances as described above.

**Hyper-Parameters and Learning Architectures.** The overall training and testing procedures closely mimic Barreto et al. [2]. The successor features $\boldsymbol{\psi}^\pi$ and their distribution $\boldsymbol{\Psi}^\pi$ are represented as multi-layer perceptrons (MLP) with two hidden layers of size 256 and $\tanh$ non-linearities. The SFC51 and RaSFC51 architectures are generally identical and require output layers of dimensions $\mathbb{R}^{9 \times 51 \times 13}$, with a $\mathrm{softmax}$ activation function applied with respect to the second dimension. Similarly, C51 and RaC51 also require output layers but of dimensions $\mathbb{R}^{9 \times 51}$ and $\mathrm{softmax}$ applied with respect to the second dimension. For SFDQN, the output of the network is linear with dimensions $\mathbb{R}^{9 \times 13}$. We also use target networks for both SFC51/RaSFC51 and C51/RaC51, which are updated every 1,000 transitions by copying weights from the learning networks. These target networks are only used for computing the bootstrapped return estimates. For SFC51, RaSFC51 and SFDQN, separate MLPs are used to learn each policy. To allow C51 and RaC51 to generalize across target locations, we apply *universal value function approximation* [35] and incorporate the target position into the state. This makes C51 essentially identical to the DQN baseline in Barreto et al. [2], except that DQN is replaced by C51. For C51-based agents, recall that the range of possible values of $\phi$ must also be specified. For SFC51 and RaSFC51, we use $\phi^d_{min} = -1$ and $\phi^d_{max} = 1$ for $d = 1, 2 \ldots 12$ and use $\phi^{13}_{min} = 0$ and $\phi^{13}_{max} = 1$, which corresponds to a relatively tight bound for state features described in the previous paragraph. For C51 and RaC51, we set the bounds to $V_{min} = -30$ and $V_{max} = 10$, which corresponds to a tight bound for the discounted return. These intervals are discretized into $N = 51$ atoms for learning histograms, as recommended in the original paper [6].

**Training and Testing Procedures.** Agents are trained on all 4 training task instances sequentially one at a time, for 200,000 time steps per task using an epsilon-greedy policy with $\varepsilon = 0.1$. Analogous to Barreto et al. [2], every sample is used to train all 4 policies simultaneously for SFC51, RaSFC51 and SFDQN. A randomized replay buffer of infinite capacity stores all previously-observed transitions $(s, a, \phi, s')$ from all 4 training tasks, to avoid "catastrophic forgetting" of previously learned task instances. Each update of the network is based on a mini-batch of size 32 sampled uniformly from the replay buffer, and uses the Adam optimizer with a learning rate of $10^{-3}$. Please note that these parameters, and those in the previous paragraph, are generally identical to those used in Barreto et al. [2]. Testing follows an epsilon-greedy policy with $\varepsilon = 0.03$ and greedy actions are selected according to risk-aware GPI, e.g. $a^* \in \arg\max_a \max_{i \in \{1, \ldots 4\}} \{\tilde{\boldsymbol{\psi}}^{\pi_i}(s, a)^\mathsf{T} \mathbf{w}_j + \beta \mathbf{w}_j^\mathsf{T} \tilde{\Sigma}^{\pi_i}(s, a) \mathbf{w}_j\}$. Recall that test rewards $\mathbf{w}_j$ are provided to the agent. We set the episode length to $T = 500$ time steps for training and testing. All visualizations are based on estimating the test return at regular intervals of 5,000 time steps, calculated as the average performance of 5 independent rollouts.

**Normalization of Returns.** Since the performance varies for different target locations, Barreto et al. [2] applies a normalization procedure to compare the performance between tasks in a fair manner. We apply the same procedure, by first training a standard C51 agent from scratch on each training and test task 10 times, and recording the average performance at the beginning and end of training, $\bar{G}_b$ and $\bar{G}_a$, respectively. The normalized return illustrated in all figures is then calculated as $G_n = (G - \bar{G}_b)/(\bar{G}_a - \bar{G}_b)$.

**Physics Simulator.** The physics simulator used for the reacher domain is provided by the open-source `pybullet` and `pybullet-gym` packages [50, 51]. We adapted the Python environment in the latter package to handle multiple target goal locations as required in our problem setting. Please note that this package is released under the MIT license.

### C.4 Additional Details for Reproducibility

**Reproducing Four-Room.** The four-room experiment was run on an Alienware m17 R3, whose software and hardware specifications are provided in Table 3. Please note that while this machine has a GPU and tensorflow installed, neither were used in this experiment.

| Component | Description | Quantity |
|---|---|---|
| Operating System | Windows 10 Home | |
| Python | 3.8.5 (Anaconda) | |
| tensorflow | 2.3.1 | |
| System Memory | 32 GB | |
| Hard Disk | 953.9GB | 1 |
| CPU | Intel i7-10875H @ 2.30GHz (turbo-boost @ 5.1GHz) | 1 |
| GPU | Nvidia RTX 2080 Super 8GB | 1 |

Table 3: Software and hardware configuration used to run all experiments for the four-room domain.

**Reproducing Reacher.** The reacher experiment was run on a Lenovo ThinkStation P920 workstation, whose software and hardware specifications are described in Table 4.

| Component | Description | Quantity |
|---|---|---|
| Operating System | Ubuntu 18.04 | |
| Python | 3.8.5 | |
| tensorflow | 2.4.0 | |
| System Memory | 187 GB | |
| Hard Disk | 953.9GB | 5 |
| CPU | Intel Xeon Gold 6234 @ 3.30GHz (turbo-boost @ 4GHz) | 32 |
| GPU | Nvidia Quadro RTX 8000 48GB | 2 |

Table 4: Software and hardware configuration used to run all experiments for the reacher domain.

**Other Factors.** Please note that seeds were not fixed during the experiment but generated in each trial using Python's default seed generation algorithm. This allows us to average the performance of all algorithms over different seed values and initializations. No internal modifications to the Python environment nor to any of its installed packages were made. No effort to overclock the machines' CPUs or GPUs beyond their factory settings were made in order to decrease the overall computation time (see below).

**Computation Time.** The majority of the computation time in running the experiment was allocated to the reacher domain, partially because of the size of the network architectures required to learn meaningful policies (2 hidden layers consisting of 256 neurons), and the number of samples required to draw meaningful conclusions for all baselines. The computation time is considerably greater for RaSFC51 (around 28-36 hours per trial) than it is for RaC51 (around 6-8 hours per trial), which is expected since the former must train 4 neural networks while the latter must train only one. This could potentially lead to negative environmental impacts if the model is to be deployed on complex problems in real-world settings. At the same time, the potential speed-ups demonstrated by RaSFC51 as compared to RaC51 could reduce the *overall* training time considerably and offset the total energy requirement of learning policies with a satisfactory variance-adjusted return. Parallelization of the training loop could also be beneficial and provide significant time and cost savings in practice.

## D   Additional Ablation Studies and Plots

In this section, we include the full details and results of the ablation studies described in the main text, and additional analysis that had to be left out of the main paper due to space limitations.

### D.1   Four-Room

**Effect of Varying $\beta$.** We can study the effect of $\beta$ on the return performance and risk-sensitivity of the learned behaviors by repeating the four-room experiment (Appendix C.2) for various values of $\beta$. In particular, we trained RaSFQL for $\beta \in \{0, -1/2, -1, -2, -4\}$ ($\omega$ for RaPRQL), and recorded

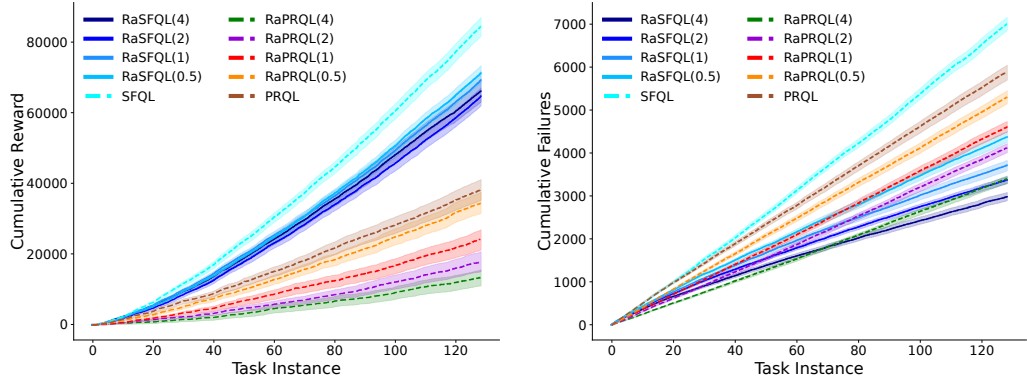

Figure 9: **Left:** cumulative reward collected across all training tasks in the four-room domain, for various values of $\beta$ for RaSFQL ($\omega$ for RaPRQL). **Right:** cumulative number of failures across all training tasks in the four-room domain, for various values of $\beta, \omega$. Please note that legend entries in parentheses indicate the negative values of $\beta$ and $\omega$. Shaded error bars indicate one standard error over 30 independent runs of each algorithm.

the cumulative reward and number of failures across all 128 training task instances. The results of these experiments are summarized in Figure 9. We see that the performance of RaSFQL degrades gracefully as $\beta$ decreases (a relative drop in cumulative reward of approximately $25\%$ is observed when $\beta$ is decreased from 0 to $-4$), while the corresponding degradation for RaPRQL is considerably more pronounced (a relative drop in cumulative reward of roughly $75\%$ is observed for an identical change in $\omega$). Meanwhile, the number of cumulative failures of RaSFQL is generally lower than RaPRQL for every pair of identical values of $\beta$ and $\omega$. In fact, for $\beta \in \{-1, -2, -4\}$, the cumulative numbers of failures are increasing at sub-linear rates, which implies that risk-avoidance behavior is becoming more prominent as the number of training task instances increases.

**Examination of Learned Behaviors.** In order to better understand the kind of risk-averse behaviors being learned, we instantiated 27 novel test task instances by enumerating $\mathbf{w}_i \in \{-1, 0, 1\}$ for every object class $i = 1, 2, 3$. We then tested the performance of the GPI policy obtained from the training procedure described in the main paper, by simulating 100 rollouts following the epsilon-greedy policy with $\varepsilon = 0.1$ on each of the test tasks. Please note that no training was ever performed on the test tasks. The state visitation counts across all 100 trajectories were computed for every task instance and arranged in a 3D-lattice as indicated in Figure 10. We repeated this procedure twice: once for RaSFQL with $\beta = -2$ and once for SFQL. Interestingly, RaSFQL and SFQL learn behaviors that are similar to each other when looking at the same task, but each of them exploits different regions of the state space depending on the reward. However, RaSFQL almost always learns to avoid the dangerous objects in the bottom-left and top-right rooms, whereas SFQL does not necessarily do so. This discrepancy is most evident, for instance, when $\mathbf{w}_2 = 1$ and for $(\mathbf{w}_1, \mathbf{w}_2, \mathbf{w}_3) = (1, -1, -1)$ and $(\mathbf{w}_1, \mathbf{w}_2, \mathbf{w}_3) = (1, -1, 0)$.

### D.2 Reacher

**Benefit of Distributional RL for Learning Successor Features.** The left plot in Figure 11 illustrates the normalized test return, averaged across all test tasks, for SFC51 and the original SFDQN implementation of Barreto et al. [2]. Both agents are risk-neutral in this comparison. It is likely that learning the full distribution of SF returns provides additional stability of the Bellman backups in stochastic domains, and thus allows SFs to inherit the advantages of distributional RL for maximizing expected return [6].

**Effect of Varying $\beta$.** Unlike the four-room domain, we saw that the number of failures of RaSFC51 was modestly *greater* than RaC51 for the same values of $\beta$. In order to better understand how efficiently the trade-off between risk and reward is handled by these two algorithms, we decided to compute an alternative measure of return by dividing the normalized return by the total number

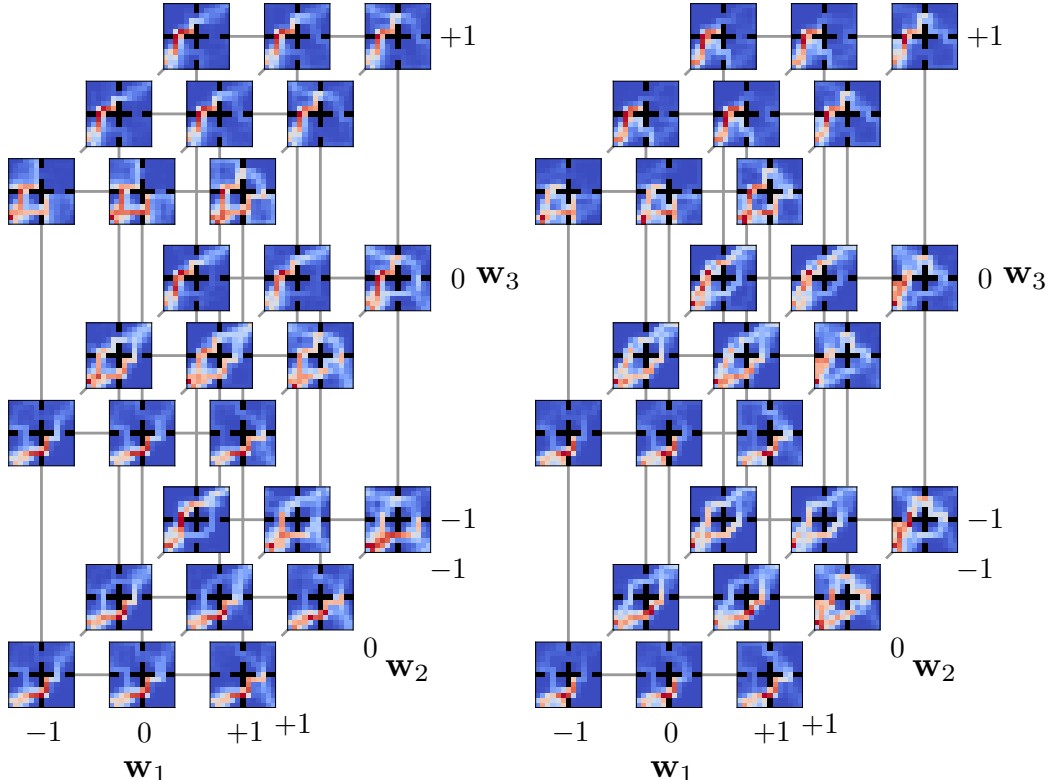

Figure 10: Visitation counts over 100 rollouts from behavior/training policies (epsilon-greedy with $\varepsilon = 0.12$) derived from GPI after training on all 128 task instances. The behavior policies are illustrated on 27 novel task instances in which the reward $\mathbf{w}$ varies, e.g. $\mathbf{w}_1, \mathbf{w}_2, \mathbf{w}_3 \in \{-1, 0, 1\}$. **Left:** Behavior policies derived from GPI for RaSFQL with $\beta = -2$. **Right:** Behavior policies derived from GPI for standard SFQL. Visitation counts are averaged over 30 independent runs for each algorithm.

of failures. Intuitively, this quantity provides an estimate of the expected reward collected between between successive failure events. The right plot contained in Figure 11, which compares this quantity for RaSFC51 and RaC51 for different values of $\beta$, shows that RaSFC51 is actually much more efficient at managing the trade-off between risk and reward for larger-magnitude values of $\beta$. This is not surprising, given that RaSFC51 can obtain much high return than RaC51 for a comparable number of failures, when $\beta = -3$ and $\beta = -4$. In fact, for $\beta = -4$, the number of failures of RaSFC51 and RaC51 become equivalent as both methods learn sufficiently conservative policies. Even in this case, successor features combined with GPI allow RaSFC51 to generalize much better on novel tasks than RaC51.

**Examination of Learned Behaviors.** As suggested in the main text, one possible conjecture is that RaSFC51 learns to correctly solve the test tasks, requiring the robotic arm to hover closer to the edge of the risky areas, while RaC51 does not. The presence of environment stochasticity, errors in function approximation, and the stochasticity of the epsilon-greedy policy used during testing could exacerbate this. Comparing rollouts of successfully-learned behavior produced by RaSFC51 and RaC51 in training tasks in Figure 12, and testing tasks in Figure 13, confirms that RaSFC51 is much better at task generalization than RaC51. Here, RaSFC51 learns to hover right at the boundaries of the high-variance regions, preferring not to enter them whenever possible. On the other hand, risk-neutral SFC51 is completely unaware of the risky areas, focusing exclusively on minimizing the distance to the target location, but is able to successfully locate all targets. RaC51 demonstrates similar risk-aware behaviors as RaSFC51, but cannot reliably locate the target on some of the test task instances.

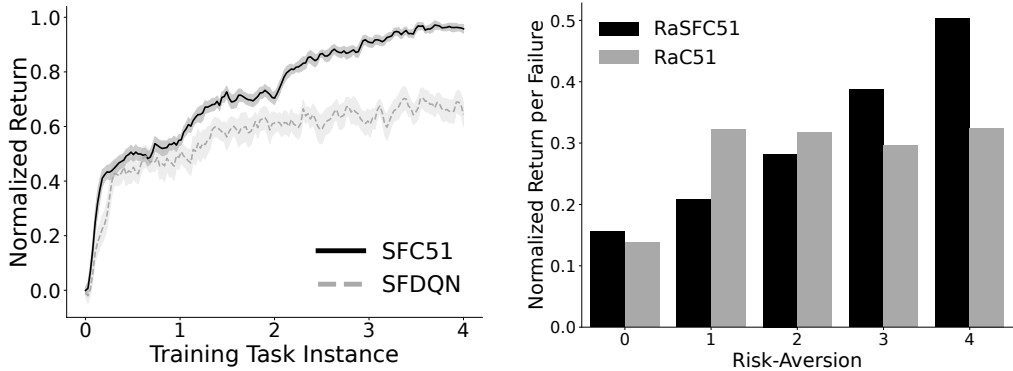

Figure 11: Additional ablation studies for the reacher domain, that were left out of the main paper due to space limitations. **Left:** Normalized average test return for the reacher domain, showing the improvement obtained by replacing DQN by C51 as a function approximator for SFs. **Right:** In order to assess the trade-off between return and possibility of failure, we divide the normalized return, averaged across all test tasks, by the total number of failures for each value of $\beta$. The resulting measure is compared between RaSFC51 and RaC51. The x-axis indicates the negative values of $\beta$.

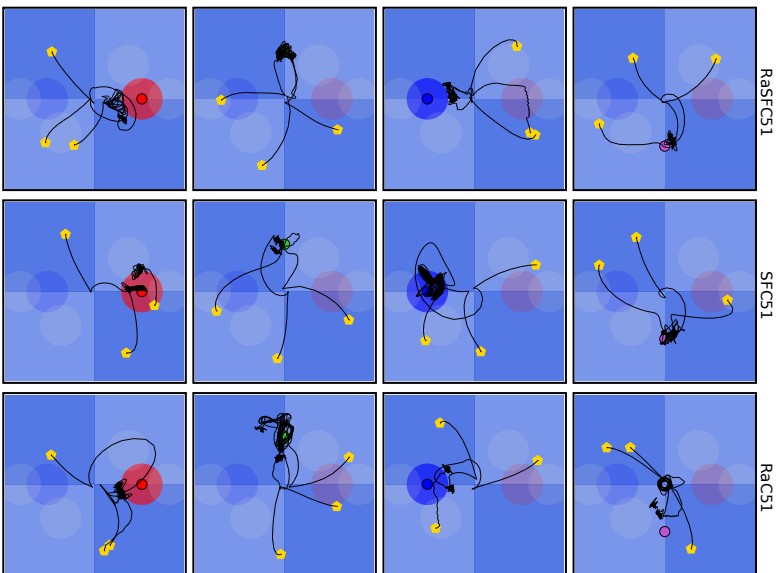

Figure 12: Evolutions of the robotic arm tip position in three successful rollouts of the reacher domain according to the GPI policy obtained after training on all 4 tasks. Here, all 4 training tasks are shown.

**Examination of Learned Covariance.** A similar conclusion can also be drawn by observing the heat-maps of the learned mean-variance objectives in Figure 15. For SFC51, these objectives take the highest values precisely at the target locations, whereas for RaSFC51 these take the highest values slightly away from the targets in regions of low volatility. This is expected as the utility of hovering very close to a target location centered in a risky region should be lower than hovering outside the risky region, for a sufficiently risk-averse agent. Moreover, the first 4 rows correspond to training task values and the last 8 correspond to test task values. Because a similar pattern described above can also be observed in test tasks, the ability of SFs to generalize expected return estimates to novel task instances also extends to higher-order sufficient statistics, namely the variance of return. Finally, the aggregated plots located in the top half in Figure 16 show that RaSFC51 learns the return variance correctly after having trained on all 4 task instances. On the other hand, the SFDQN architecture that learns the covariance using the residual method (8) is unable to learn the variance correctly, likely due to the propagation of errors and overestimation bias in $\tilde{\psi}^{\pi_i}(s, a)$ as discussed in the main paper.

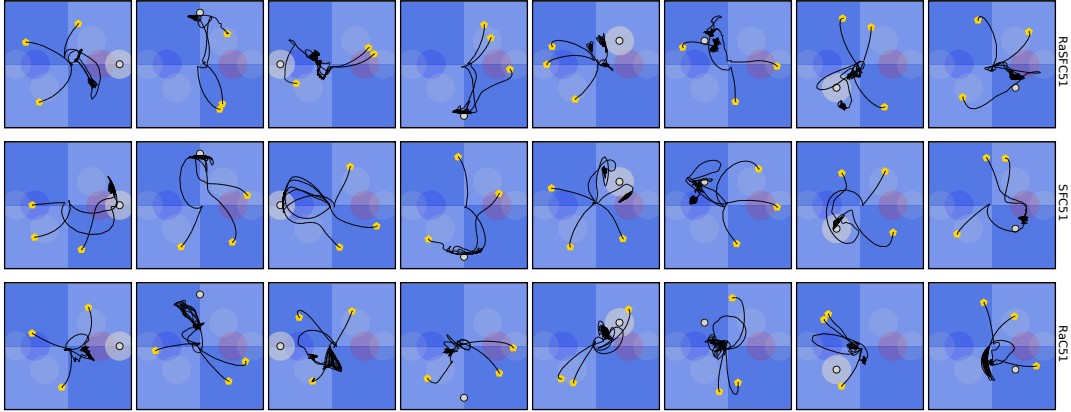

Figure 13: Evolutions of the robotic arm tip position in three successful rollouts of the reacher domain according to the GPI policy obtained after training on all 4 tasks. Here, all 8 test tasks are shown.

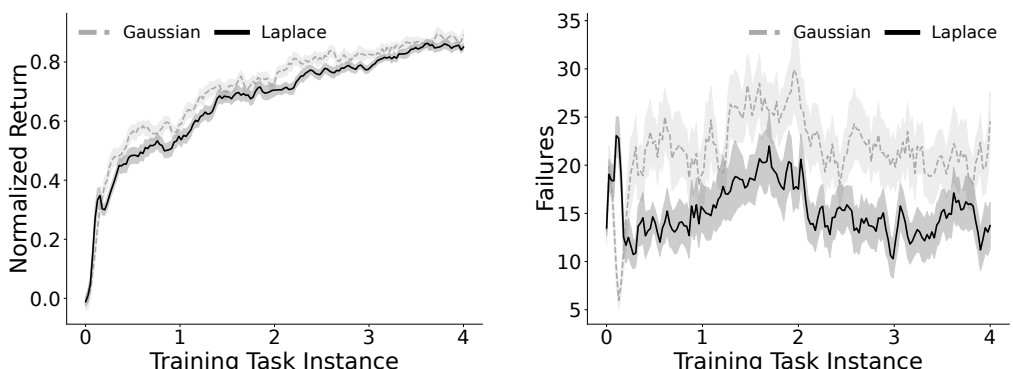

Figure 14: Ablation study for varying the distributional assumption used in the mean-variance approximation in the reacher domain. Gaussian implies the usual mean-variance approximation with the assumption of normally-distributed returns, while Laplace assumes a Laplace distribution for returns. Both approximations use $\beta = -2$. **Left:** Normalized average test return across all test tasks. **Right:** Total failures across all test tasks.

**Effect of Varying the Distributional Assumption of Returns.** A final ablation study we conducted was to assess the stability of the task generalization of risk-aware SFs under different distributional assumptions of the cumulative reward. As discussed in Section A.4, the mean-variance approximation used in this work results from making a Gaussian assumption on the cumulative reward distribution, which may not always be closely-aligned with reality. To assess the flexibility of our approach, we evaluate the performance of RaSFC51 when the distribution of cumulative reward is assumed to follow a Laplace approximation (see Table 2 for details). We repeat the reacher experiment with this modification and report the results (test return and test failures) in Figure 14. As we can see, the normalized test return under both distributional assumptions is essentially unchanged. However, the number of test failures is reduced to almost half its original amount by simply replacing the normal distribution with the Laplace distribution for modeling returns. By inspecting and comparing the penalties of these two distributional assumptions in Figure 8, this result is expected, since the Laplace distribution is better-suited for modeling heavy-tailed risks and thus penalizes the variance more than the normal distribution. This further suggests that the assumptions used for modeling the distributions of returns in GPE can have a profound effect on the effectiveness of transfer and the risk-sensitivity of the learned policies. Thus, these distributional assumptions can – and perhaps should – be treated as important hyper-parameters or design parameters in future experiments. Designing and characterizing suitable distributional assumptions for different classes of problems could be an interesting topic for further investigation.

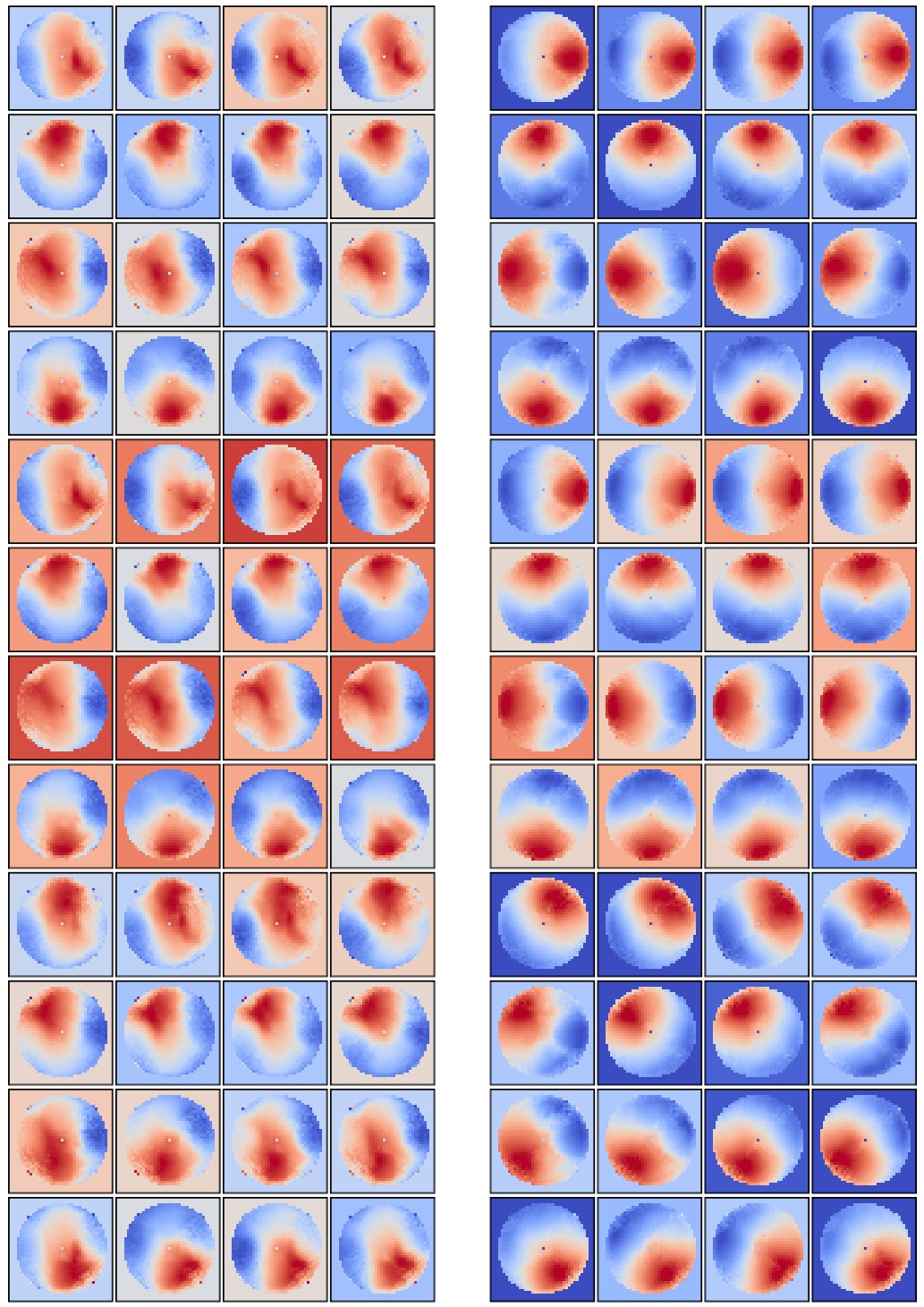

Figure 15: Each plot located in column $i$ and row $j$ illustrates the value of the mean-variance objective $\boldsymbol{\psi}^{\pi_i}(s,a)^\mathsf{T}\mathbf{w}_j - \mathbf{w}_j^\mathsf{T}\Sigma^{\pi_i}(s,a)\mathbf{w}_j$ as a function of the robotic arm tip position in $(x,y)$ coordinates for the reacher domain, after training each agent on all 4 tasks. In other words, the first 4 rows illustrate the value functions learned on the training task instances, while the last 8 rows illustrate the value functions learned on the test tasks. **Left:** mean-variance objective computed by RaSFC51 with $\beta = -3$. **Right:** mean-variance objective computed by SFC51.

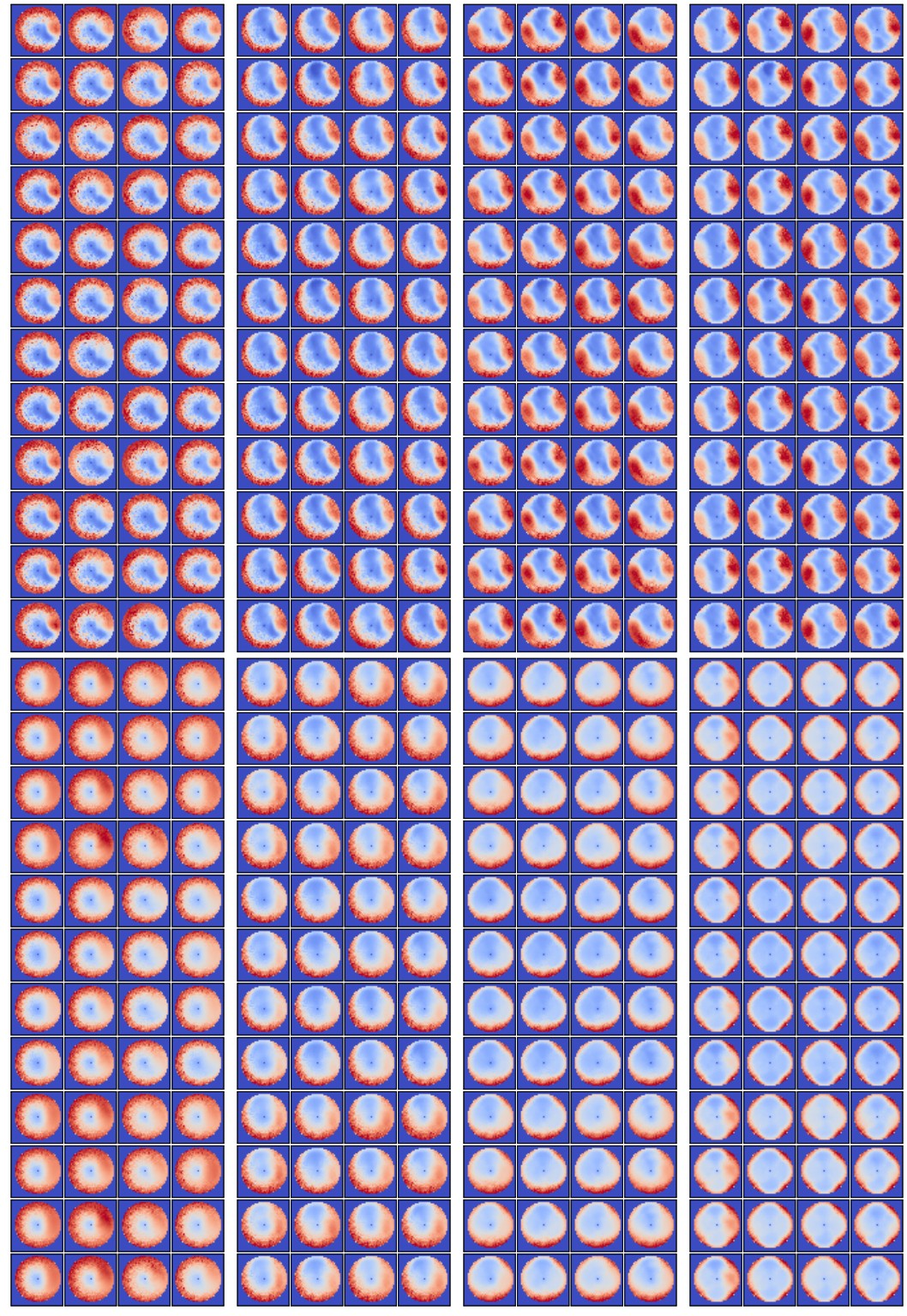

Figure 16: Each plot located in column $i$ and row $j$ illustrates the value of the variance $\mathbf{w}_j{}^\intercal\Sigma^{\pi_i}(s,a)\mathbf{w}_j$ as a function of the robotic arm tip position in $(x,y)$ coordinates for the reacher domain, after training on 1, 2, 3 and 4 source tasks (respectively, left to right). **Top:** variance computed by RaSFC51 with $\beta = -3$. **Bottom:** variance computed by SFDQN using (8).