# OpenReview forum: "Risk-Aware Transfer in Reinforcement Learning using Successor Features"
_NeurIPS.cc/2021/Conference — NeurIPS 2021 Poster_

### Official Review · Reviewer_TAaE · 2021-07-03

**Rating:** 6
**Confidence:** 2

**Summary:**

In this paper, a (variance-based) risk-aware transfer method for multi-task RL is proposed.
In the theory part, GPI and GPE (and the notion of successor features) are extended to entropic utility (for the risk-aware transfer).
Based on the results in the theory part (specifically, the GPE/successor features parts), the successor feature for entropic utility is proposed to use for risk-aware setting.
In the experiment part, the proposed method is evaluated in the Four-room and Reacher environments.


**Limitations And Societal Impact:**

Yes

**Main Review:**

Key strength:
Solid theoretical results are provided.

Theories of (vanilla) GPI and GPE (Theories 1 and 2 in [2]) are extended to entropic utility (i.e., Theories 1 and 2 in the paper under review). Although the derivation (proof) flow for the theories is basically similar to those for the vanilla GPI and GPE in [2], the theoretical results are meaningful because the results extend the GPI and GPE into a risk-aware setting.
In addition, Theorem 3 (convergence of covariance) is novel.

Key weakness:
Evaluation of the proposed method (RaSF) is conducted only in a simple environment.
In the evaluation, Four-Room (grid-world) and Reacher (simple control) are used for benchmark environments.
It is unclear if it works well in more complex environments (e.g., Ant-maze with traps).


I will improve my score if a convincing rebuttal to my comments on "Key weakness" is provided (e.g., the additional experiment result in more complex environments).


Minor comments:
Line 76 Other previous work on safe options: Takuya Hiraoka, Takahisa Imagawa, Tatsuya Mori, Takashi Onishi, Yoshimasa Tsuruoka. Learning Robust Options by Conditional Value at Risk Optimization, In NeurIPS, 2019
Line 431 Deep rl -> Deep RL
Line 442 NeurIPS -> In NeurIPS


**Time Spent Reviewing:**

10

---

> ### Author Response · Authors · 2021-08-10
> **Response to Reviewer TAaE**
>
> Thank you for your detailed feedback and the additional reference. We answer your questions and concerns below:
>
> > “Evaluation of the proposed method (RaSF) is conducted only in a simple environment”
>
> Indeed, it is a natural question to ask how well our framework will generalize to more complex tasks. We are currently looking into running the proposed experiment (Ant-maze with traps) and adding it to the final version of the paper. We note that GPE and GPI have been successfully used in the Ant domain [see, e.g. Barreto et al., 2019], so we are optimistic that their risk-aware counterparts should work there as well.
>
> The main reasoning for the choice of gridworld and reacher was to ensure fair comparison with [2], where they use the same experimental setup but without traps or risky states. This allows us to show that our risk-aware successor feature representation inherits all the advantages of the risk-neutral counterpart, while also taking risk into account. Also, please note that these tasks are much harder to solve than the original domains in [2], because the noise in the state and reward leads to slower learning, and learning correct risk-sensitive policies is often harder than learning risk-neutral ones. This is where incorporating risk in the objective proves to be particularly powerful, since we can avoid risky regions.
>
> More generally, we believe that solving highly complex domains in this framework is possible by leveraging advances in learning representations on which our practical implementation is based. Particularly, we see two ways to improve the RaSF framework in the future:
> 1.  Distributional RL: It was shown to learn much better representations on complex tasks such as Atari as shown in prior work (see Figure 6, center for further verification of this, where we compared to regular SFs). Since SFs are essentially value functions, the guarantees offered by distributional RL also extend naturally to our setting (please see responses to reviewer TjXA regarding interaction effects and how this can be addressed in an elegant, tractable and theoretically justified manner).
> 2.  Advances in SFs: Secondly, as you have correctly noted, our theory is particularly meaningful for risk, since we can offer the same theoretical guarantees as regular SFs while taking variance into account. Since SFs were already successful in solving complex visual tasks, and can be paired with LSTMs and CNNs (see, e.g. [3]), this is also possible in our framework and is expected to improve the way in which reward and risk are represented. This is yet another interesting direction for improving the performance of RaSFs while maintaining the theoretical guarantees that are critical in our paper for ensuring that these improvements are meaningful.
>
> Barreto, A., Borsa, D., Hou, S., Comanici, G., Aygün, E., Hamel, P., Toyama, D., Hunt, J., Mourad, S., Silver, D. and Precup, D., 2019, December. The option keyboard combining skills in reinforcement learning. In Proceedings of the 33rd International Conference on Neural Information Processing Systems (pp. 13052-13062).

---

> > ### Comment · Reviewer_TAaE · 2021-08-26
> > **Response to authors**
> >
> > I appreciate the author's response.
> >
> > I somewhat agree with the authors' positive view on the application of the proposed method in more challenging environments. However, the actual results were not provided during this discussion period, so I would like to keep the score as it is (I am still leaning to recommend accept).

---

> > > ### Author Response · Authors · 2021-08-29
> > > **Thank you for your suggestions**
> > >
> > > Thank you very much for your detailed comments. We will focus on the additional experiment in the ant domain for the final version of the paper, introducing stochastic transitions and traps into the dynamics as we have done for the reacher domain. On a more general level, and as noted in our response, our method is following the framework of GPE and GPI quite closely and leverages modern deep RL methodology (eg. distributional RL). Thus, we believe that our method is well-suited for solving problems of similar complexity (or even greater, as already shown empirically on reacher) as those which risk-neutral successor features are able to solve.

---

### Official Review · Reviewer_TjXA · 2021-07-17

**Rating:** 7
**Confidence:** 5

**Summary:**

The article extends the Successor Feature framework to the risk-averse setting in which the Entropic Risk Measure is optimized in place of the standard risk-neutral objective. The proposed approach is tested on a gridworld domain and on a Mujoco one, this time integrated with a distributional RL algorithm.

**Limitations And Societal Impact:**

No potential negative impact are present.

**Main Review:**

## Originality:
The approach is a direct extension of successor features, based on the properties of the Bellman equation of the entropic risk-measure and on its Taylor expansion. Related works are clearly cited in a dedicated section.


## Quality:
The submission is technically sound from both the theoretical and the empirical point of view. The author determine the theoretical properties of the proposed approach an then tested them in valid environments.
From the experimental side, I only have doubt about the choice of the performance measure. The author chose to show the expected return, together with the failures. However, it would have been fairer to show the optimized objective (the entropic risk measure). The latter may also suffer of instability issues, hence, it would have been interesting to see if this aspects affect the proposed technique.

## Clarity:
The paper is well written and organized in a adequate way. Both the experimental analysis and the theory are sufficiently clear to understand the illustrated results. I would suggest anyway minor improvements on some specific parts of the article:
- the example in 3.1, can be exposed in a clearer way, in particular, it is not clear what happens when \beta is equal to 0: which are the optimal policies in the different tasks, in this case?
- In Theorem 1, it is not clear, without looking at the proof, what is the difference between Q and \Tilde{Q}.
- in Figure 3, it is not immediate to understand the performance of the different techniques, maybe a table would do a better job.
- In Figure 4, I would avoid to use faded curves, they are difficult to read. Also, in the second figure, the x-label should be "Test task Instance", from what I understand from the caption.
- In the "Reacher" paragraph of the experimental section, it is not clear what "without interactions effect" means.
- Showing the variance instead of the total failures is also an interesting alternative, justified by the fact that ERM approximate mean-variance, and the studied setting is not a CMDP.

## Significance:
The proposed approach outperforms the baselines in the first environment, but not its risk-neutral version, even if it allows for a lower number of failures. A similar behavior is observed in the second environment. It is hence difficult to assess whether the proposed approach is better then its risk-neutral version, if the final perfomance w.r.t. the optimization goal is not shown. Nevertheless, the approach offers an interesting an theoretically justified way to transfer learning in risk-averse tasks, representing a clear advance in the state of the art.

**Time Spent Reviewing:**

3

---

> ### Author Response · Authors · 2021-08-10
> **Response to Reviewer TjXA**
>
> > “what happens when \beta is equal to 0: which are the optimal policies in the different tasks, in this case?”
>
> In the case where beta = 0, the agent should learn risk-neutral policies only. This is predicted by the theory, since for beta = 0 we recover the original GPE/GPI formulation, and has also been observed in practice. Specifically, upon running this experiment, we found that all learned policies try to get to the goal using the shortest route possible. This indeed makes it clear that risk-sensitivity is a prerequisite for learning safe behavior in this example, a good point that we will clarify in Section 3.1.
>
> > “what is the difference between Q and \Tilde{Q}”
>
> Indeed, we will rewrite the statement to clarify that Q is the true utility and \tilde(Q) is the estimate. Thank you for pointing this out.
>
> > “ it is not clear what "without interactions effect" means.”
>
> This means that we estimate the marginal distributions when calculating variance rather than the full joint distribution, that is, we do not take into account all the interactions between features. We will add this explanation to the text to make this point clearer. By the way, the reason we did not take all the interactions into account is that this would quickly become intractable as the feature dimensionality d grows.
>
> One way to alleviate this is by estimating the joint distribution of the SFs implicitly through quantiles (see, e.g. Dabney et al., 2018). However, this is well outside the scope of this work, because it would be necessary to derive a suitable loss function (e.g. Wasserstein) and a learning algorithm for minimizing it with some form of guarantees; both of these are non-trivial questions. We believe our work will motivate the need to develop a principled framework for high-dimensional distributional RL, and our work will benefit from it.
>
> > “Showing the variance instead of the total failures is also an interesting alternative”
>
> We agree that showing the mean-variance or entropic objective will be insightful, as it will ensure that the successor feature formulation maintains high values when generalizing to novel reward instances. In a sense, Figures 14 and 15 in the appendix do show the correct variance-adjusted values are obtained through generalization. However, we can certainly add these plots, either to the main paper if space permits, or the appendix.

---

> > ### Comment · Reviewer_TjXA · 2021-08-18
> > **Answer to authors**
> >
> > Thank you for addressing my doubts, I will increase my score.

---

> > > ### Author Response · Authors · 2021-08-29
> > > **Thank you for your comments**
> > >
> > > Thank you very much for your comments and suggestions, and revision. We will address your concerns about the notation in the theorems, improve the clarity of the empirical results and the more relevant objectives for comparison in the camera-ready.

---

### Official Review · Reviewer_5osE · 2021-07-19

**Rating:** 6
**Confidence:** 4

**Summary:**

This paper extends the successor feature framework including the GPE/GPI methods for generalisation across tasks with risk awareness. The paper includes experiments for the associated methods on toy domains.

**Limitations And Societal Impact:**

As mentioned before the main limitation of the work is lack of evidence to support existence and learnability of succinct SFs to describe the space of practical risk sensitive problem domains. Additionally the theoretical results presented do not appear very informative for generalization.

**Main Review:**

Overall the paper has good clarity, however the theoretical results do not seem super useful and the the paper simultaneously lacks on strong empirical evaluation.

The authors should try to address the following:
- How is sample inefficiency a cause for not addressing safety, lines 21-24 are bit unclear.
- A little derivation for taylor expansion in line 98 should be provided in main text
- How useful are Theorem 1 and 2? Ideally one would want to have little uncertainty in the initial Q estimates for transfer as the policy objective is to maximise expected value under initial state distribution, however the theorems gives a very weak bound for h=0
- The method not only assumes access to SFs, which while being problematic to compute themselves also requires estimates for additional quantities like their covariance matrix, I feel this pushes most of the burden of solving hard tasks under the carpet. It is not very convincing to me that tasks which are interesting to solve and pose real challenges with respect to risk uncertainties would simultaneously admit low dimensional SF descriptors (note this is really important for the method given computability of O(d^2) structures like covariance matrix, d being SF dimensionality). Further the lack of thorough experimentation on this front compounds my doubts.
- Are there guarantees which extend for general \beta for the GPI case? Currently it seems that the source and the target tasks assume same risk awareness coefficient.

Updates post discussion:
- I think the authors have made convincing points to address some of the issues raised in my review, also since they will be adding more experiments as stated, I have therefore decided to increase my rating for acceptance.



**Time Spent Reviewing:**

7

---

> ### Author Response · Authors · 2021-08-10
> **Response to Reviewer 5osE**
>
> Thank you for your very detailed comments. We provide our answers to your questions below:
>
> > “How is sample inefficiency a cause for not addressing safety?”
>
> The line of reasoning in lines 21-24 does not claim that addressing sample inefficiency will also address safety. On the contrary, sample inefficiency is only one deficit of modern RL that must be addressed in order to allow us to deploy algorithms more safely and reliably in real world settings, and a principal way to do this is through transfer learning. However, while transfer learning is a promising way to address learning efficiency and improve policy quality, it does not guarantee that the resulting policies will be sensitive to risk. The novelty of our work is precisely to provide a mechanism for transferring risk-aware policies between tasks while preserving their sensitivity to risk. As far as we know, this is the first method that allows one to do so in a principled way.
>
> > “How useful are Theorem 1 and 2?”
>
> We believe that Theorems 1 and 2 are critical in order to ensure monotone transfer and asymptotic convergence of GPI policies when risk of the MDPs is taken into account. Previously, this had only been done in the risk-neutral setting [2], whereas incorporating risk is one of the distinguishing features of our work. In fact, our work is a strict generalization of [2] that reduces to the risk neutral case only when \beta = 0. As such, we do not believe that sharper bounds can be proved without making more restrictive assumptions on the MDPs.
>
> Furthermore, we would like to point out that your doubts about the usefulness of the theory are not specific to our proposed method, but are related fundamentally to GPI and GPE in the risk-neutral setting as well. To see this, please consider that:
> 1. Theorems 1 and 2 are looking at the undiscounted finite horizon setting, where the value error (the left side of the inequalities) measures the discrepancy in total accumulated risk-adjusted reward between tasks. In fact, dividing both sides of these equations by (T – h + 1), the average error in Q-values, amortized per decision epoch, now becomes bounded uniformly by epsilon for all h. We expect that identical bounds hold in the risk neutral setting as well, though not proved before [2].
> 2.  The corresponding results for the discounted setting, proved in Appendix B.3, are just as strong as the episodic case, where the error is a discounted sum of \epsilon, e.g.  $\epsilon/(1-\gamma)$. Indeed, these error bounds correspond exactly to the risk-neutral setting [2] for \beta = 0.
>
> > “The method not only assumes access to SFs, which while being problematic to compute themselves also requires estimates for additional quantities like their covariance matrix”
>
> We agree that not all tasks may have a natural SF representation. However, we argue that this is not directly related to the problem we are trying to address in our paper, namely, risk-aware transfer.  Note that the existence or not of a compact SF representation is an issue that also arises in the risk-neutral setting, and as such it has been extensively discussed and dealt with in the literature. For an example, we redirect the reviewer to [3], where it is shown that a good set of SFs can always be constructed by using the reward functions as basis.
>
> It is true that the risk-aware formulation turns a O(d) method into a O(d^2) counterpart. This is indeed specific to our approach. We argue that this is a relatively small price to pay to be able to handle risk, especially in scenarios where failures are not an option.  Please also see comments to reviewer TjXA on how this can be addressed more elegantly in future work, without explicitly depending on the feature dimension d.
>
> > “Are there guarantees which extend for general \beta for the GPI case? “
>
> This is a very good point. While our current results assume that \beta remains fixed among source and target tasks, we believe it is possible to apply our framework in some sense when \beta changes. To see this, please note that Theorem 1 did not place any assumptions on the source policies, making it possible to build a policy library with different values of \beta, while still ensuring that the GPI policy improves monotonically w.r.t. the target task. On the other hand, the consequences of Theorem 2 are harder to show, since the optimal policies with respect to a given value of beta will not, in general, be optimal for other values of beta. The GPI policy will still improve with every single policy added, though, so in the limit it should perform optimally under any beta.
>
> From the perspective of GPE, we note that the mean-variance approximation can be written in closed-form as a function of beta (as well as w, as in the original formulation). Given the sufficient statistics (successor features and covariance) for a source policy allows the policy to be evaluated on demand for arbitrary \beta and arbitrary w. We should point out that such generalization to risk levels is not possible in previous work.
>
> Transfer with varying levels of \beta is indeed a very interesting application of our current framework, since we would expect that long-lived decision makers want to adjust the risk-level in response to feedback from the environment in an iterative manner. Another interesting use case is when different users want to deploy policies on demand for various risk levels, such as providing risk tolerance levels to an autonomous vehicle together with a destination. It would be interesting to see whether, given a reward vector w, GPI learns to assemble a meaningful policy from the source policy library that respects the desired (or approximate) level of risk aversion. Thank you for this excellent question, which we will discuss in the revision.

---

> > ### Comment · Reviewer_5osE · 2021-08-31
> > **Discussion Phase**
> >
> > The author response seems promising in partially addressing some of my queries (general \beta for GPI).
> >
> > However, I am not totally convinced that the amortization of the transfer error bound would really address the problem of getting optimal expected returns on the new task (as it is taken w.r.t. start state distribution with h=0). If the authors can provide discussion/results on how tight these bounds are for the given SF framework, that would be interesting.
> >
> > Further, additional experiments on somewhat more complex environments should be provided (as also raised by other reviewers). Since the authors seem to use approximations for SF covariance matrix to get around computational intractability, I think it is important to test this at scale for larger/harder tasks.
> >
> > I will keep my scores for now but would be willing to change them provided the authors address the above issues in the revision/rebuttal as they suggest.

---

> > > ### Author Response · Authors · 2021-09-02
> > > **Tightness of GPI/GPE bounds in total reward setting**
> > >
> > > Thank you for your comment about the tightness of the bounds and sharing your concerns. To alleviate this concern you may have, we decided to derive a counterexample to show that the bounds are indeed tight and cannot be reduced without further information.
> > >
> > > > " If the authors can provide discussion/results on how tight these bounds are for the given SF framework, that would be interesting."
> > >
> > > To this end, consider a simple MDP with two actions $a_0$ and $a_1$ and two states $s_0$ and $s_1$, with dynamics defined by $P(s_0 | s_i, a_0) = P(s_1 | s_i, a_1) = 1$ for all $i = 0, 1$ (in other words, whichever action I take today entirely determines the next state). Furthermore, define the features $\phi_i(s) = 1[s = s_i]$ for $i = 0, 1$ as one hot encoding and the parameterized reward $\mathbf{w} = [w_1, w_2]$. We will ignore the approximation errors and assume the values are optimally estimated, e.g. $\varepsilon = 0$.
> > >
> > > For task number 1 we will define $\mathbf{w}_1 = [1, 0]$ and the optimal policy $\pi_1^*$ always chooses action $a_0$ in each state. The optimal value function for task 1 in horizon $T$ (e.g. with $T$ steps left in the episode) becomes:
> > > \begin{equation}
> > >  Q_1^{\pi_1^*}(s_0, a_0) = T, \quad
> > > Q_1^{\pi_1^*}(s_1, a_0) = T - 1, \quad
> > > Q_1^{\pi_1^*}(s_0, a_1) = T - 1, \quad
> > > Q_1^{\pi_1^*}(s_1, a_1) = T - 2
> > > \end{equation}
> > > Similarly, for task 2 we will define $\mathbf{w}_2 = [0, 1]$ and the optimal policy $\pi_2^*$ always selects action $a_1$ in each state. The corresponding value function for task 2 becomes
> > > \begin{equation}
> > >  Q_2^{\pi_2^*}(s_0, a_0) = T - 2, \quad
> > > Q_2^{\pi_2^*}(s_1, a_0) = T - 1, \quad
> > > Q_2^{\pi_2^*}(s_0, a_1) = T - 1, \quad
> > > Q_2^{\pi_2^*}(s_1, a_1) = T
> > > \end{equation}
> > >
> > > To demonstrate that GPI attains the upper bound, we will apply GPI + GPE with the policy $\pi_1^*$ to solve task 2. For example, starting in pair $s_0, a_0$ and following $\pi_1^*$ thereafter, we can see that $Q_2^{\pi_1^*}(s_0, a_0) = 0$, since the agent never visits $s_1$ thereafter. The gap is clearly
> > > \begin{equation}
> > > Q_2^{\pi_2^*}(s_0, a_0) - Q_2^{\pi_1^*}(s_0, a_0) = (T - 2).
> > > \end{equation}
> > >
> > > Thus the gap grows linearly in the length of horizon $T$, and is a factor 2 away from the theoretical upper bound guaranteed by GPI since $\sup_{s}|\mathbf{\phi}(s)'\mathbf{w}_2 - \mathbf{\phi}(s)'\mathbf{w}_1| = 1$. While this result is expected to hold for risk-neutral GPI, our results are much stronger in that we expect the risk-aware GPI/GPE bounds to hold for the entropic utility for *all* possible risk aversion parameters $\beta$. Since our results correspond to regular GPI/GPE when $\beta = 0$, this counterexample applies to our setting just as well, and so the bound we prove cannot be improved without placing further assumptions on the class of MDPs we wish to solve. Our results, therefore, preserve the optimality of regular successor features up to at most a constant factor of 2, but for all choices of risk aversion.
> > >
> > > Also, please note that GPI/GPE can also be optimal in our setting, for instance in an MDP with only a single feasible policy.
> > >
> > > > "I think it is important to test this at scale for larger/harder tasks."
> > >
> > > We agree with this statement and we are working on additional experiments for the camera ready submission.

---

### Official Review · Reviewer_cpAq · 2021-07-22

**Rating:** 7
**Confidence:** 4

**Summary:**

This paper considers safety in transfer RL, and proposes a method, called Risk-aware successor features, that exploits the task structure to avoid risks and achieve task generalization between tasks with shared dynamics and different goals. The entropic utility is introduced to represent risk-awareness and specialized to entropic Q-value with bellman update. Risk-aware GPI and GPE are extended from GPI, GPE, and proved its convergence. Experiments on two domains show the method outperforms previous works.

**Limitations And Societal Impact:**

The authors only discuss some limitations in the conclusion.

**Main Review:**

This paper presents Risk-aware Successor Features (RaSFs) for realizing policy transfer between tasks with shared dynamics and different goals. The objective is to optimize the trade-off between expected return and risk, which is measured by the variance of return.

Strengths:

1. This paper extends risk-aware GPI, provides monotone guarantees and optimality of GPI, extends GPE to the mean-variance objective, inheriting the superior task generalization abilities of successor features.
2. The analysis on both discrete and continuous domains has showed.
3. The paper is clear to read, easy to follow.

Weaknesses:

Why PRQL(2005) is a challenging baseline? Why not consider more recent policy transfer methods?

Typos:
Line269: The w are also learned --> is

**Time Spent Reviewing:**

12

---

> ### Author Response · Authors · 2021-08-10
> **Response to Reviewer cpAq**
>
> Thank you for your feedback. Please see our response to your good question below:
>
> > "Why PRQL(2005) is a challenging baseline?"
>
> There are several reasons for choosing PRQL as a baseline in this work. First, we wanted to follow the experimental design of Barreto et al., 2017 [2] as closely as possible, where they also use risk-neutral PRQL in the tabular setting, citing it as a strong baseline in their evaluation. Since our work leverages the successor feature representation to transfer risk sensitive policies we believe making PRQL risk sensitive was a very natural choice. It is also meaningful to incorporate risk via variance estimates in PRQL, making it possible to isolate/ablate the benefits of using successor features as a knowledge representation while optimizing the same mean-variance objective (this baseline with variance estimates is closer to Gehring and Precup, 2013, cited in our paper).
>
> > "The authors only discuss some limitations in the conclusion."
>
> We will do our best to address other limitations, including those mentioned in the reviews as well as a few that were left out due to space limitations.

---

### Author Response · Authors · 2021-08-29
**Gentle reminder**

Dear Reviewers,

We would like to thank you for your detailed suggestions regarding added experiments, baselines and presentation, which we are currently working on to incorporate into the final version of our paper. As we are nearing the end of the discussion phase, please let us know if you have any further suggestions you believe should be incorporated into the revision, or if you have any further questions about our paper. Thanks!

---

### Decision · Program_Chairs · 2021-09-27

**Decision:**

Accept (Poster)

**Comment:**

After reading each other's reviews and the authors' feedback, the reviewers discussed the merits and flaws of the paper.
The authors' feedback was instrumental in the reviewers' change of mind.
The paper is still borderline, but I trust the authors' promises to revise their paper according to the reviewers' suggestions. and I propose to accept this paper.
I will check the final version of this paper hoping to find all the additions that have been requested.